# De novo design of high-affinity binders of bioactive helical peptides

Susana Vázquez Torres[1,2,3,12], Philip J. Y. Leung[1,2,4,12], Preetham Venkatesh[1,2,3,12], Isaac D. Lutz[1,2,5], Fabian Hink[6], Huu-Hien Huynh[7], Jessica Becker[7], Andy Hsien-Wei Yeh[1,2], David Juergens[1,2,4], Nathaniel R. Bennett[1,2,4], Andrew N. Hoofnagle[7], Eric Huang[8], Michael J. MacCoss[8], Marc Expòsit[1,2,4], Gyu Rie Lee[1,2], Asim K. Bera[1,2], Alex Kang[1,2], Joshmyn De La Cruz[1,2], Paul M. Levine[1,2], Xinting Li[1,2], Mila Lamb[1,2], Stacey R. Gerben[1,2], Analisa Murray[1,2], Piper Heine[1,2], Elif Nihal Korkmaz[1,2], Jeff Nivala[9,10], Lance Stewart[1,2], Joseph L. Watson[1,2✉], Joseph M. Rogers[6✉] & David Baker[1,2,11✉]

Many peptide hormones form an α-helix on binding their receptors[1–4], and sensitive methods for their detection could contribute to better clinical management of disease[5]. De novo protein design can now generate binders with high affinity and specificity to structured proteins[6,7]. However, the design of interactions between proteins and short peptides with helical propensity is an unmet challenge. Here we describe parametric generation and deep learning-based methods for designing proteins to address this challenge. We show that by extending RFdiffusion[8] to enable binder design to flexible targets, and to refining input structure models by successive noising and denoising (partial diffusion), picomolar-affinity binders can be generated to helical peptide targets by either refining designs generated with other methods, or completely de novo starting from random noise distributions without any subsequent experimental optimization. The RFdiffusion designs enable the enrichment and subsequent detection of parathyroid hormone and glucagon by mass spectrometry, and the construction of bioluminescence-based protein biosensors. The ability to design binders to conformationally variable targets, and to optimize by partial diffusion both natural and designed proteins, should be broadly useful.

Peptide hormones, such as parathyroid hormone (PTH), neuropeptide Y (NPY), glucagon (GCG) and secretin (SCT), which adopt α-helical structures on binding their receptors[1–4], play key roles in human biology and are well-established biomarkers in clinical care and biomedical research (Fig. 1a). There is considerable interest in their sensitive and specific quantification[9], which at present relies on antibodies that require substantial resources to generate, can be difficult to produce with high affinity, and often have less-than-desirable stability and reproducibility[10–14]. The loop-mediated interaction surfaces of antibodies are not particularly well suited to high-specificity binding of extended helical peptides—almost all anti-peptide antibodies bind their targets in non-helical conformations[15]. Designed proteins with very high stability can be readily produced with high yield and low cost in *Escherichia coli*; however, although there have been considerable advances in de novo design of binders for folded proteins[6,7], the design of proteins that bind helical peptides with high affinity and specificity remains an outstanding challenge. Design of peptide-binding proteins is challenging for two reasons.

First, proteins designed to bind folded proteins, such as picomolar-affinity hyper-stable 50–65-residue minibinders[7,16], have shapes suitable for binding rigid concave targets, but not for cradling extended peptides. Helical peptides can readily associate to form coiled-coil assemblies, and this principle has been used to design binders for a calmodulin peptide[17], but coiled-coil subunits generally self-associate in the absence of binding partners owing to considerable exposed hydrophobic surface, considerably reducing the effective target-binding affinity. Second, peptides have fewer residues to interact with, and are often partially or entirely unstructured in isolation[18]. Hence, there can be an entropic cost of structuring the peptide into a specific conformation[19], which compromises the favourable free energy of association. Progress has been made in designing peptides that bind to extended β-strand structures[20] and polyproline II conformations[21] using protein side chains to interact with the peptide backbone, but such interactions cannot be made with α-helical peptides owing to the extensive internal backbone–backbone hydrogen bonding.

[1]Department of Biochemistry, University of Washington, Seattle, WA, USA. [2]Institute for Protein Design, University of Washington, Seattle, WA, USA. [3]Graduate Program in Biological Physics, Structure and Design, University of Washington, Seattle, WA, USA. [4]Graduate Program in Molecular Engineering, University of Washington, Seattle, WA, USA. [5]Department of Bioengineering, University of Washington, Seattle, WA, USA. [6]Department of Drug Design and Pharmacology, University of Copenhagen, Copenhagen, Denmark. [7]Department of Laboratory Medicine and Pathology, University of Washington, Seattle, WA, USA. [8]Department of Genome Sciences, University of Washington, Seattle, WA, USA. [9]School of Computer Science and Engineering, University of Washington, Seattle, WA, USA. [10]Molecular Engineering and Sciences Institute, University of Washington, Seattle, WA, USA. [11]Howard Hughes Medical Institute, University of Washington, Seattle, WA, USA. [12]These authors contributed equally: Susana Vázquez Torres, Philip J. Y. Leung, Preetham Venkatesh. ✉e-mail: jwatson3@uw.edu; joseph.rogers@sund.ku.dk; dabaker@uw.edu

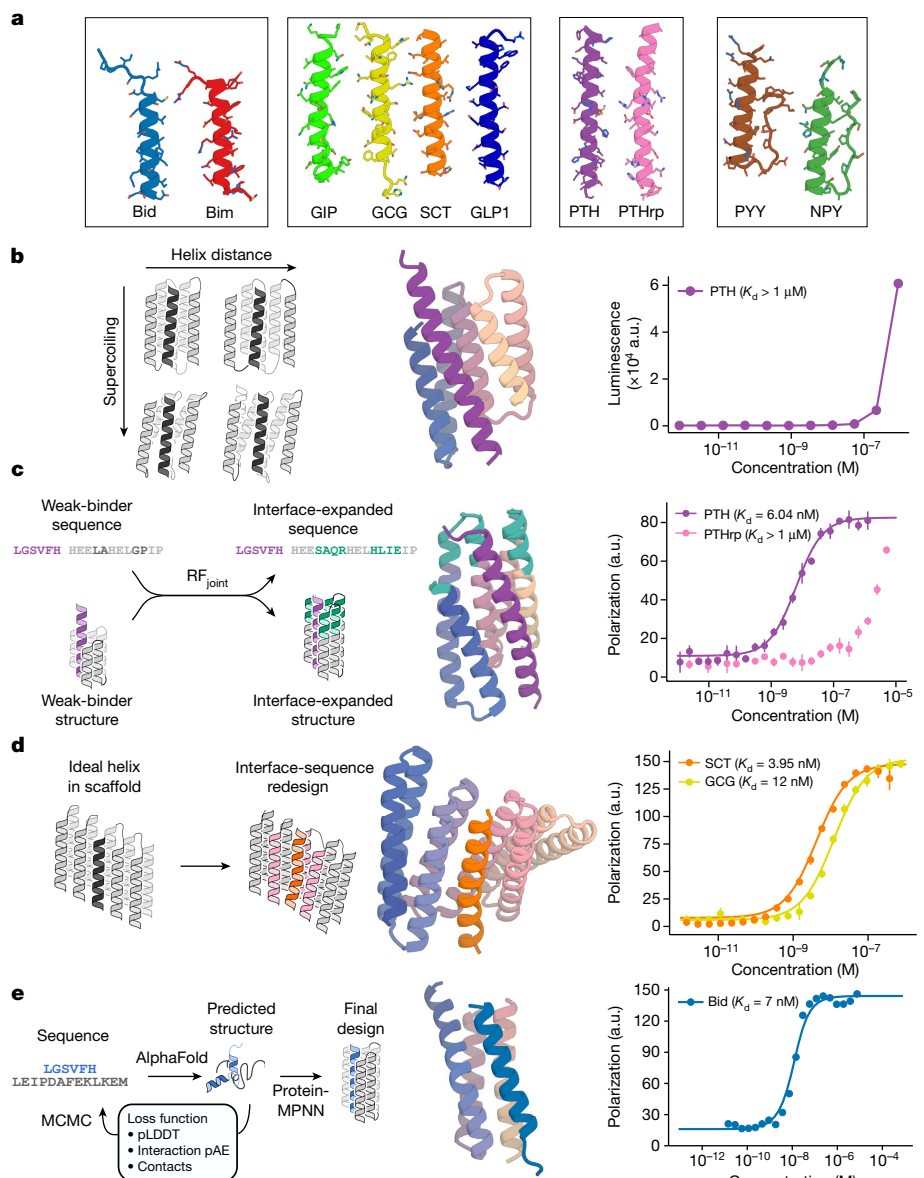

**Fig. 1 | Design strategies for binding helical peptides. a**, Helical peptide targets: apoptosis-related BH3 domains of Bid[43] (PDB ID: 4QVE) and Bim[44] (PDB ID: 3FDL), GCG[3] (PDB ID: 1GCN), gastric inhibitory peptide[45] (GIP; PDB ID: 2QKH), SCT[46] (PDB ID: 6WZG), GCG-like peptide 1[47] (GLP1; PDB ID: 6X18), PTH[48] (PDB ID: 1ET1), PTHrp[25] (PDB ID: 7VVJ), PYY[49] (PDB ID: 2DEZ) and NPY[50] (PDB ID: 7X9A). **b**, Parametric approach. Left: sampling groove scaffolds varying supercoiling and helix distance to fit different targets. Middle: design model (spectrum) and PTH target (purple) of the best parametrically designed PTH binder. Right: split NanoBiT titration of PTH and the binder showed weak binding. a.u., arbitrary units. **c**, Inpainting binder optimization. Left: redesign of parametrically generated binder designs using RF$_{joint}$ Inpainting to expand the binding interface and ProteinMPNN to redesign the sequences. Middle: AF2 prediction of Inpainted design (spectrum) with extended interface (teal), and PTH target (purple). Right: FP measurements ($n = 4$) indicate 6.04 nM binding to PTH and weak binding to off-target PTHrp. **d**, Threading approach to peptide binder design. Left: starting with a helix-bound scaffold, a target is threaded onto the bound helix and the interface is redesigned. Middle: AF2 prediction of design (spectrum) and SCT target (orange). Right: FP measurements ($n = 4$) indicate 3.95 nM binding to SCT and 12 nM binding to GCG. **e**, Hallucinating peptide binders. Left: Markov chain Monte Carlo (MCMC) steps are carried out in sequence space. At each step, the peptide sequence is re-predicted, and changes are accepted or rejected on the basis of interfacial contacts and AF2 metrics. The final structure is then redesigned using ProteinMPNN to avoid adversarial sequences. Interaction pAE, predicted alignment error across the interface; pLDDT, predicted local distance difference test. Middle: AF2 prediction of design (spectrum) and Bid target (blue). Right: FP measurements ($n = 4$) indicate 7 nM binding affinity to Bid.

## Design of peptide-binding scaffolds

We set out to develop general methods for designing proteins that bind peptides in helical conformations. To fully leverage recent advances in protein design, we explored both parametric and deep learning-based approaches. For parametric generation, we reasoned that helical bundle scaffolds with an open groove for a helical peptide could provide a general solution to the helical peptide-binding problem: the extended interaction surface between the full length of the helical peptide target and the contacting helices on the designed scaffold could enable high-affinity and specific binding, and the helices flanking the groove could limit self-association of the recessed hydrophobic surfaces. In parallel, we reasoned that deep learning methods, which do not pre-specify scaffold geometries, could permit the exploration of different potential solutions to peptide binding.

We began by exploring parametric methods for generating backbones with overall 'groove' shapes. Using the Crick parameterization of α-helical coiled coils[22], we devised a method to sample scaffolds consisting of a three-helix groove supported by two buttressing helices (Fig. 1b and Methods). We assembled a library sampling a range of supercoiling and helix–helix spacings to accommodate a variety of helical peptide targets (Supplementary Figs. 1–3). We then used this library to design binders to PTH, GCG and NPY, and screened 12 designs for each target using a NanoBiT split-luciferase binding assay (Supplementary Fig. 4). Many of the designs bound their targets (3, 4 and 8 out of 12 to PTH, GCG and NPY, respectively) but with only micromolar affinities (Fig. 1b and Supplementary Fig. 4a–c). These results suggest that groove-shaped scaffolds can be designed to bind helical peptides, but also that design method improvement was necessary to achieve high-affinity binding.

We next explored using RoseTTAFold Inpainting ($RF_{joint}$)[23], a model that can jointly design protein sequences and structures, along with ProteinMPNN[24], an improved sequence design method, to improve the modest affinity of our tightest parametrically designed PTH binder (Fig. 1c, left). We used $RF_{joint}$ Inpainting to extend the binder interfaces and ProteinMPNN to redesign the sequences, reasoning that the combination of these two methods could lead to more favourable interactions with the peptide. Out of 192 designs tested, 44 showed binding against PTH in initial yeast display screening. Following size-exclusion chromatography (SEC), the best binder was found to bind with 6.04 nM affinity to PTH using fluorescence polarization (FP). Binding was specific: very little binding was observed to PTH-related peptide (PTHrp), a related peptide sequence with 34% sequence identity that binds the same receptor as PTH[25] (Fig. 1c, right). Overall, the affinity of the starting PTH binder was improved by approximately three orders of magnitude, and the computational model of the highest-affinity binder had 19% greater surface area contacting the target peptide (the structural extension was critical to the improvement in binding affinity; sequence redesign with ProteinMPNN of the original binding interface did not measurably increase affinity; Supplementary Fig. 5). We used the same design strategy to generate higher-affinity binders for NPY and GCG. Using weak parametric binders as a starting point, we extended their binding interfaces and redesigned their sequences to generate a 231-nM-affinity binder for GCG and a 3.5-μM binder for NPY after screening 96 designs (Extended Data Fig. 1a,b).

As an alternative to de novo parametric design of scaffolds that contain grooves, we explored the threading of helical peptides of interest onto already existing designed scaffolds with interfaces that make extensive interactions with helical peptides[26,27] (Fig. 1d, left, and Methods). We threaded sequences of peptides of interest onto these complexes, and filtered for interfacial hydrophobic interactions between the target sequence and the scaffolds[17,26]. The selected scaffolds were then redesigned in the presence of the threaded target sequence with ProteinMPNN[24] and the complex was predicted with AlphaFold2[28] (AF2; with initial guess[6]) and filtered on AF2 and Rosetta metrics. Initial screening using yeast surface display identified 4/66 binders for SCT, which were expressed in *E. coli*. After purification, all four of the designs were found to bind with submicromolar affinity using FP, with the highest-affinity design binding with an affinity of 2.7 nM for SCT (Fig. 1d, right); we also made designs with a dissociation constant ($K_d$) of <100 nM to GCG-like peptide 1 and gastric inhibitory polypeptide (Extended Data Fig. 2a,b). The SCT binder design bound GCG, which has 44% sequence identity to SCT[4,29], with fourfold weaker affinity than SCT (Fig. 1d, right).

## Designing binders using Hallucination

We next explored the use of deep learning Hallucination methods to generate helical peptide binders completely de novo, with no pre-specification of the binder or peptide geometry (Fig. 1e, left inset, and Supplementary Fig. 6a). Hallucination or 'activation maximization'

approaches start from a network that predicts protein structure from sequence, and carry out an optimization in sequence space for sequences that fold to structures with desired properties. This approach has been used to generate new monomers[30], functional-site scaffolds[23] and cyclic oligomers[31]. Hallucination using AF2 or RoseTTAFold has the advantage that neither the binder nor the peptide structure needs to be specified during the design process, enabling the design of binders to peptides in different conformations (this is useful given the unstructured nature of many peptides in solution; disordered peptides can bind in different conformations to different binding partners[18]). Hallucination directly optimizes metrics correlated with binding, albeit with the possible hazard of generating adversarial protein sequences[31]. We began by designing binders to the apoptosis-related BH3 domain of Bid (Fig. 1a). The Bid peptide is unstructured in isolation, but adopts an α-helix on binding to Bcl-2 family members[32,33]; it is therefore a model candidate for the design of helix-binding proteins. Starting from only the Bid primary sequence, and a random seed binder sequence (of length 60, 70, 80, 90 or 100 residues), we carried out a Monte Carlo search in sequence space, optimizing for confident binding to the target peptide (AF2 predicted local distance difference test, pLDDT; and predicted alignment error, pAE)[6]. The trajectories typically converged in 5,000 steps (sequence substitutions; Supplementary Fig. 6b), and the output binder structure was subsequently redesigned with ProteinMPNN, as previously described[31]. All designed binders were predicted to bind to Bid in a predominantly helical conformation; the exact conformations differ between designs because only the amino acid sequence of the target is specified in advance. This protocol effectively carries out flexible backbone protein design, which can be a challenge for traditional Rosetta-based design approaches for which deep conformational sampling can be very compute intensive. In line with our prediction that 'groove' scaffolds would offer an ideal topology for helical peptide binding, many of the binders from this approach contain a well-defined 'groove', with the peptide predicted to make extensive interactions with the binder, typically helix–helix interactions (Extended Data Fig. 3a).

We experimentally tested 46 of the Hallucinated designs (Extended Data Fig. 3a) by co-expression of a GFP-tagged Bid peptide and His-tagged binders, with co-elution of GFP and binder used as a read-out for binding. Four of these designs were further characterized, and showed soluble, monomeric expression even in the absence of peptide co-expression (Extended Data Fig. 3b), and could be pulled down using Bid BH3 peptide immobilized on beads (Extended Data Fig. 3c). Circular dichroism experiments indicated that the Bid peptide was unstructured in solution, and that helicity increased on interaction with the Hallucinated proteins, in line with the design prediction (Extended Data Fig. 3d). The binders were highly thermostable and, unlike the native Bcl-2 protein Mcl-1, readily refolded after (partial) thermal denaturation at 95 °C (Extended Data Fig. 3e). FP measurements revealed a 7-nM-affinity binder to Bid peptide (Fig. 1e, right inset), a higher-affinity interaction than that between Bid and the native partner Mcl-1 (Extended Data Fig. 3f,g).

## Design refinement with RFdiffusion

We next explored using the RoseTTAFold-based denoising diffusion method RFdiffusion[8]. RFdiffusion directly generates protein structures with diverse topologies, and is much more compute efficient than Hallucination. We first extended RFdiffusion to enable optimization of existing helical peptide-binders.

A long-standing challenge in protein design is to increase the activity of an input native protein or designed protein by exploring the space of plausible closely related conformations for those with predicted higher activity[34]. This is difficult for traditional design methods as extensive full-atom calculations are needed for each sample around a starting structure (using molecular dynamics simulation or Rosetta full-atom relaxation methods), and it is not straightforward to optimize

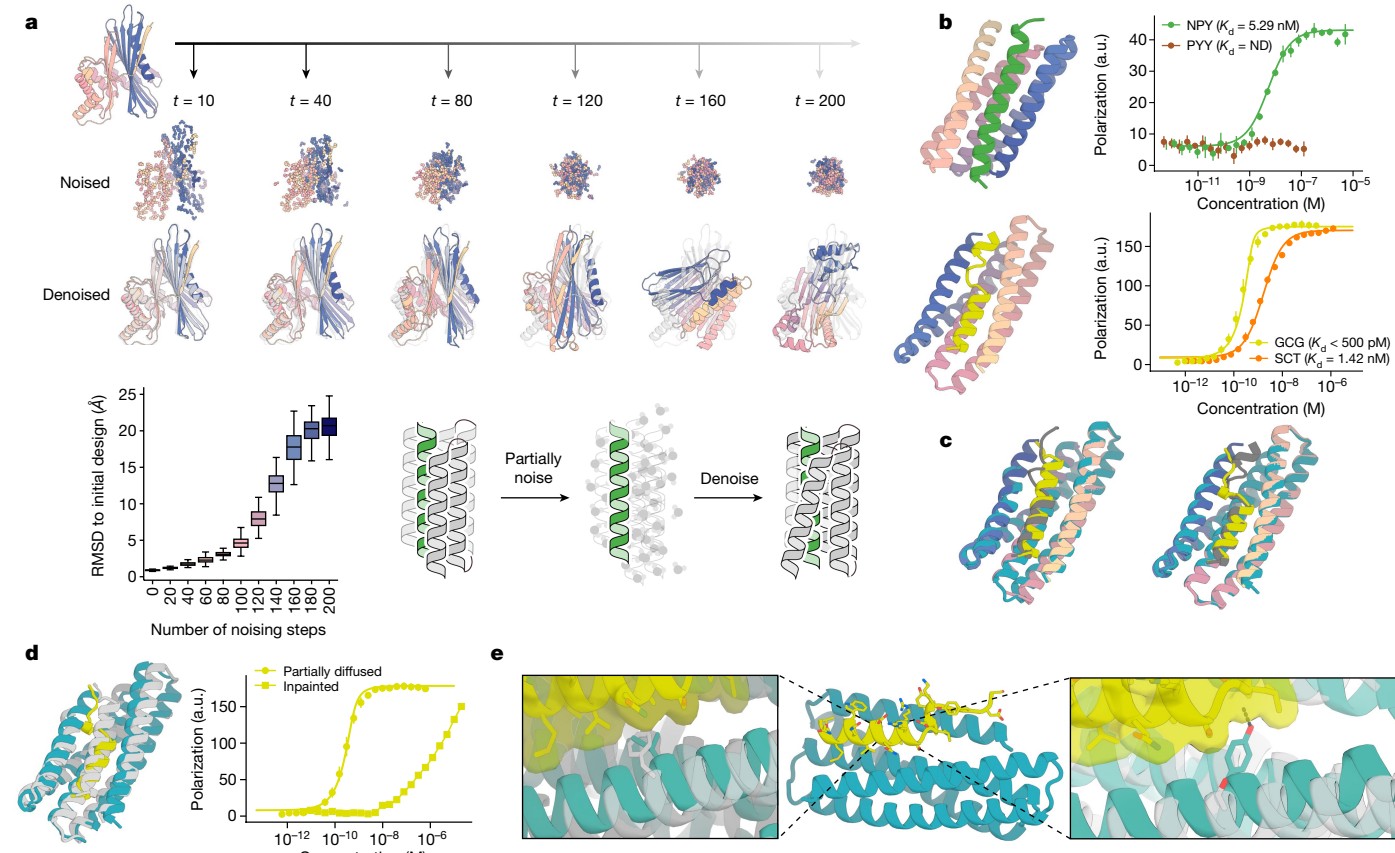

**Fig. 2 | Peptide binder optimization with RFdiffusion. a**, Top: partial diffusion. RFdiffusion is used to denoise a randomly noised starting design (left); varying the extent of initial noising (middle row) enables control over the extent of introduced structural variation (bottom row; colours, new designs; grey, original design). Bottom left: partial diffusion diversifies designs. Note that the greater the amount of noise added, the more dissimilar the outputs are to the starting structure. Bottom right: depiction of the helix binder optimization strategy. **b**, Top: design model (spectrum) of the partially diffused binder to NPY (green) and FP measurements ($n = 4$) indicating a 5.3 nM binding affinity to NPY target and selectivity over PYY (brown). ND, not detectable. Bottom: design model (spectrum) of the partially diffused binder to GCG (yellow) and FP measurements ($n = 4$) indicating a subnanomolar binding affinity to GCG and selectivity over SCT (orange). **c**, Left: model (spectrum with GCG in grey) aligns with 0.72 Å RMSD to the 1.95-Å crystal structure (teal and

yellow) of the RF$_{joint}$ Inpainted GCG binder. Right: model (spectrum with GCG in grey) aligns with 0.6 Å RMSD to the 1.81-Å crystal structure (teal and yellow) of the partially diffused GCG binder. **d**, Left: the crystal structures of the Inpainted (grey) and partially diffused (teal and yellow) GCG binders have considerable topological similarity; there are many small readjustments. Right: FP titrations ($n = 4$) with GCG indicate much tighter binding following partial diffusion. **e**, Left inset: the crystal structure of the partially diffused backbone (teal) shows how the newly introduced Ile13 increases shape complementarity compared to the phenylalanine in the Inpainted binder (crystal structure in grey; structures aligned on residues 16–29 of GCG). Middle: crystal structure of the partially diffused GCG binder (teal and yellow). Right inset: the backbone shifts in the partially diffused structure (teal) enable Tyr16 to make packing and hydrogen-bonding interactions with the peptide; Ser16 in the original design did not make any peptide contacts (grey).

for higher binding affinity without detailed modelling of the binder–target side-chain interactions. We reasoned that, by contrast, RFdiffusion might be able to rapidly generate plausible backbones in the vicinity of a target structure, increasing the extent and quality of interaction with the target guided by the extensive knowledge of protein structure inherent in RoseTTAFold. Typically, during the reverse diffusion (generative) process, RFdiffusion takes random Gaussian noise as input, and iteratively refines this to a new protein structure over many ($T$) steps (generally 200). Part way through this denoising process, the evolving structure no longer resembles 'pure noise', instead resembling a 'noisy' version of the final structure. We therefore reasoned that ensembles of structure with varying extents of deviation from an input structure could be generated by partially noising initial starting structures to different extents (for example, time step 70), and then denoising to a similar, but not identical, final structure (Fig. 2a; in this case, the input coordinates to RFdiffusion at time step 70 are from a noised starting structure, rather than a partially denoised random distribution).

We implemented this 'partial diffusion' approach (Methods), and sought first to assess the extent to which protein structures could

be resampled and refined with partial diffusion. As expected, partial diffusion allowed diversification of a starting protein fold, and the magnitude of this diversity could be tuned by varying how much noise was added to a starting structure (Fig. 2a). We next explored the ability of partial diffusion to 'regularize' native protein backbones using as a metric AF2 structure prediction from a single sequence. We found that RFdiffusion improves the 'designability' of protein backbones: ProteinMPNN sequence design on partially diffused native backbones (with high similarity to the native fold, Extended Data Fig. 4a,c, middle row) have improved structure recapitulation (self-consistency) by AF2 compared to both the native sequence (Extended Data Fig. 4b, pink versus grey, and Extended Data Fig. 4c, bottom row) and ProteinMPNN sequences generated from the original native backbone (Extended Data Fig. 4b, blue, and Extended Data Fig. 4c, top row). Further, we found in tests on the well-studied colicin–immunity protein system[35] that the small changes in protein backbone that partial diffusion can sample are sufficient to mediate specificity changes within protein families (Supplementary Fig. 7). Thus, partial diffusion enables protein backbone resampling and refinement, the extent of which

can be tuned by varying the amount of noise added. Furthermore, partial diffusion can considerably increase the designability of input protein models.

As a first experimental test of partial diffusion, we started from our parametrically designed Inpainted binders to GCG (with 231 nM $K_d$) and NPY (with 3.5 μM affinity; Extended Data Fig. 1a,b). Following partial noising and denoising, we identified diverse designs (Supplementary Fig. 8) that in silico had substantially improved computational metrics compared to the starting design (Supplementary Fig. 9). We used an auxiliary potential during the denoising trajectory[8] that minimized the radius of gyration (see Code availability) of the protein–peptide complex to promote additional interaction with the peptide. Initial screening with yeast surface display revealed quite high binding success rates, with 25 out of 96 designs binding GCG, and 20 out of 96 binding NPY at 10 nM peptide concentration. The highest-affinity designs were expressed in *E. coli* and then purified, and their binding affinities were determined using FP to be 5.6 nM to NPY (Fig. 2b, top) and below the limit of detection in the picomolar range to GCG (Fig. 2b, bottom). The designs were quite specific: the GCG binder bound preferentially to GCG over the closely related SCT, and particularly notably, the NPY binder did not show any cross-reactivity to peptide YY (PYY), a member of the NPY and pancreatic polypeptide family[36] with 63% sequence identity to NPY (Fig. 2b).

To gain insight into the structural rearrangements generated by partial diffusion that contribute to the affinity increases, we solved the structures (Extended Data Table 2) of the original Inpainted GCG binder and the partially diffused higher-affinity GCG binder. Both designs were very close to their design models. Subtle structural changes in the protein backbone between the original Inpainted design model and the partially diffused model are nearly perfectly recapitulated in the corresponding crystal structures (Fig. 2c). Alignment of the two crystal structures (Fig. 2d) on the structurally conserved C-terminal residues (16–29) of GCG (Supplementary Fig. 10) showed that in the partially diffused GCG binder a 2.7-Å shift towards the target in the binder backbone enables an isoleucine to fit into a pocket previously occupied by a phenylalanine side chain at position 13 (Fig. 2e, left inset). Similarly, at position 16, a 3.6-Å shift in the backbone allows a tyrosine residue to pack underneath the peptide and form a hydrogen bond to the peptide backbone where previously a serine could not make any contacts (Fig. 2e, right inset). These backbone movements and accompanying sequence changes increase the interaction shape complementarity (0.62 versus 0.67) and contact molecular surface (431 Å$^2$ versus 522 Å$^2$; computed on the crystal structures). We observed similar improvements in estimated binding energy (Rosetta ddG) and contact molecular surface after running partial diffusion starting from the Inpainted designs for GCG and NPY (Supplementary Fig. 11).

## De novo binder design using RFdiffusion

Inspired by this success at optimizing binders with RFdiffusion, we next tested its ability to design binders completely de novo through unconditional binder design. We first used the fixed target structure approach of ref. 8, and provided RFdiffusion with the sequence and structures of the two peptides in helical conformations, leaving the topology of the binding protein and the binding mode completely unspecified (Fig. 3a). From this minimal starting information, RFdiffusion generated designs predicted by AF2 to fold and bind to the targets with high in silico success rates. A representative design trajectory is shown for PTH in Supplementary Video 1; starting from a random distribution of residues surrounding the PTH peptide in a helical conformation, in sequential denoising steps the residue distribution shifts to surround the peptide and progressively organizes into a folded structure that cradles almost the entire surface of the peptide.

We obtained synthetic genes encoding 96 designs for each target. Using yeast surface display, we found that 56 of the 96 designs bound

to PTH at 10 nM peptide concentration. The highest-affinity design again bound too tightly for accurate $K_d$ estimation; instead FP data provide an approximate upper bound for the $K_d$ of < 500 pM (Fig. 3b, bottom). Binding was also highly specific; no binding was observed to the related PTHrp (Fig. 3b, bottom). For Bim, 25/96 of the designs bound by yeast surface display, and FP on the highest-affinity design indicated a $K_d$ of < 500 pM (Fig. 3c, bottom). Circular dichroism temperature melts indicate that both binders are stable at 95 °C (middle panels of Fig. 3b,c). The completely de novo diffused binders again had considerable structural similarity to our starting groove binding concept (compare the top panels of Fig. 3b,c to the middle panel of Fig. 1b). We solved the X-ray crystal structure (Extended Data Table 2) of the Bim binder, and found that it closely matched the design model (3.0 Å resolution, 0.57 Å RMSD; Fig. 3d). A kinked helix on the binder adjacent to the interface is well recapitulated in the structure, and a cross-interface hydrogen-bond network designed between Thr73 and Asn77 of the binder and Asn20 of Bim forms in the otherwise hydrophobic interface.

We next sought to generalize RFdiffusion to enable binding to flexible targets from a specification of the target sequence alone (as can be achieved with AF2 Hallucination, detailed above). We fine-tuned RFdiffusion by training on two chain systems from the Protein Data Bank (PDB), noising the structure on one and providing only the sequence on the second. We found that the fine-tuned version could readily design folded structures around a variety of peptides given only sequence information. We used this approach to design binders to PYY (Fig. 3e), which in the cryogenic electron microscopy structure with the NPY Y2 receptor is incompletely resolved and adopts a partially helical structure[37]. Starting from only the amino acid sequence of PYY, RFdiffusion generated solutions with the peptide in a range of conformations. A design with the peptide adopting a different conformation from the experimental structure bound PYY with 24.5 nM affinity (Fig. 3e, right). Note that here we explored using shorter binder chain lengths in these calculations, resulting in smaller designs, which probably accounts for the lower affinity than in the fixed structure case above. Lower affinity binders were also obtained for PTH and GCG using this flexible backbone RFdiffusion approach (Extended Data Fig. 5a,b).

## Human versus machine problem solving

The deep learning methods largely converged on the overall solution to the helical peptide-binding design problem—groove-shaped scaffolds with helices lining the binding site—that the human designers chose in the initial Rosetta parametric approaches. The increased affinity of the deep learning designs probably derives at least in part from higher shape complementarity resulting from direct building of the scaffold to match the peptide shape; the average contact molecular surface for the partially diffused GCG binders and NPY increased by 33% and 29% respectively compared to that of the starting models, and the Rosetta ddG improved by 29% and 21% (Supplementary Fig. 11). The ability of RFdiffusion de novo design to rapidly 'build to fit' provides a general route to creating high-shape-complementary binders to a wide range of target structures, and as noted above, partial diffusion provides a general route to sampling binders with increased affinity by making small backbone adjustments to enable placement of more space-filling side chains.

## Design of protein biosensors

Given our success in generating de novo binders to clinically relevant helical peptides, we next sought to test their use as detection tools for use in diagnostic assays. Compared to immunosensors, de novo protein-based biosensors can offer a more robust platform with high stability and tunability for diagnostics[38]. To design PTH biosensors,

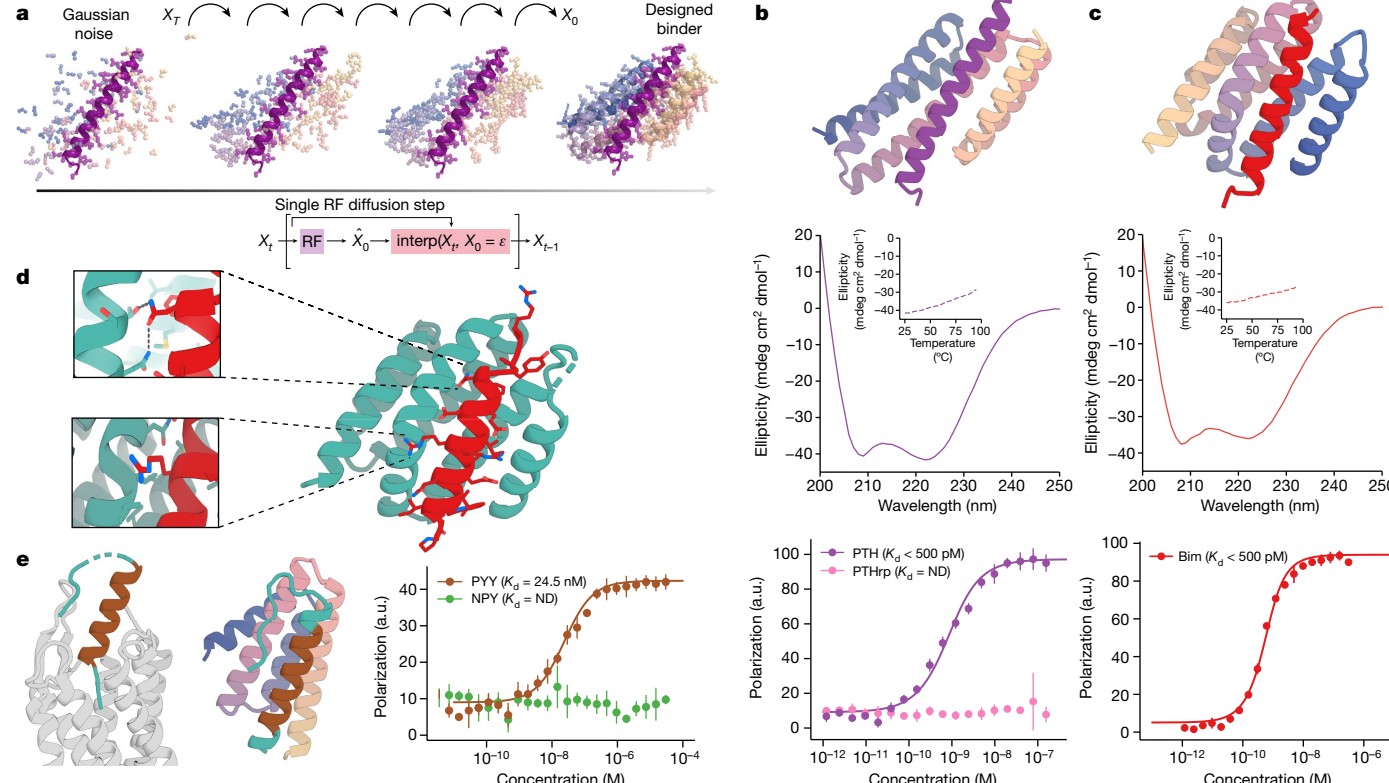

**Fig. 3 | De novo peptide binder design with RFdiffusion. a**, Schematic showing peptide binder design using RFdiffusion. Starting from a random distribution of residues around the target peptide ($X_T$), successive RFdiffusion denoising steps progressively remove the noise leading at the end of the trajectory to a folded structure, $X_0$, cradling the peptide. At each step $t$, RFdiffusion predicts the final structure p$X_0$ given the current noise sample $X_t$, and a step that interpolates in this direction is taken to generate the input for the next denoising step $X_{t-1}$. **b**, Design of picomolar-affinity PTH binder. Top: design model of PTH binder (spectrum, AF2 metrics in Supplementary Table 9). Middle: circular dichroism data show that the binder has helical secondary structure and is stable at 95 °C (inset). Bottom: FP measurements ($n = 4$) with PTH indicate a subnanomolar binding affinity and no binding to PTHrp indicates high specificity. **c**, Design of picomolar-affinity Bim binder. Top: design model of Bim binder (spectrum, AF2 metrics in Supplementary Table 9). Middle: circular dichroism data show that the binder has helical secondary structure and is stable at 95 °C (inset). Bottom: FP measurements ($n = 4$) with Bim indicate a subnanomolar binding affinity. **d**, Crystal structure of Bim binder (teal and red). Top inset: a cross-interface hydrogen-bond network formed between Asn20 of Bim and Thr73 and Asn77 of the binder. Bottom inset: a kinked helix in the diffused backbone accommodates Arg13 of Bim. **e**, RFdiffusion with PYY sequence input alone. Left: PYY in complex with its native NPY Y2 receptor[37] (PDB ID: 7YON) shows flexibility at its N and C termini (teal). Middle: design model of the binder (spectrum) with PYY target (brown); the peptide is more ordered in both regions (N terminus, teal). Right: FP measurements ($n = 4$) with PYY indicate a 24.5 nM binding affinity.

we grafted the 6.1-nM PTH binder into the lucCage system[39] (Fig. 4a), screened eight designs for their luminescence response in the presence of PTH, and identified a sensitive lucCagePTH biosensor (LOD = 10 nM) with ≈21-fold luminescence activation in the presence of PTH (Fig. 4b).

## Enrichment for LC–MS/MS detection

We explored the use of our picomolar-affinity RFdiffusion-generated binders to PTH and GCG as capture reagents in immunoaffinity enrichment coupled with liquid chromatography–tandem mass spectrometry (LC–MS/MS), a powerful platform for detecting low-abundance protein biomarkers in human serum[40]. We prepared PTH- and GCG-binder-conjugated beads as described in the Methods. PTH enrichment was quantified on the basis of the analysis of the amino-terminal peptide of a tryptic digestion of PTH in human plasma[41] (Methods and Extended Data Fig. 6a). We found that the designed binder enabled capture of PTH from buffer and human plasma supplemented with PTH (the endogenous levels are too low for reliable detection) with recoveries of 53% and 43%, respectively (Fig. 4d, left). For GCG, enrichment was quantified for recovery of peptide in buffer solution (see Methods and Extended Data Fig. 6b) because recovery was low in extract (further increases in specificity of this design will probably be necessary for

actual applications). The GCG binder beads had comparable peptide capture efficiency to that of monoclonal GCG antibody beads, with 91.1% recovery when normalized to the antibody's 100% recovery rate in a spiked buffer (Fig. 4d, right). In contrast to the antibody-coupled beads, which lost almost all GCG-binding activity after the first use (Fig. 4d, right), the GCG-binder-conjugated beads retained almost full binding activity in a second capture experiment (Fig. 4d, right). This greater robustness to washing and repeated use probably reflects the exceptional stability of the designed binders (middle panels of Fig. 3b,c and Extended Data Fig. 3e), which could substantially lower cost (as they are no longer single use) and extend shelf life compared to using antibodies.

## Discussion

Antibodies have served as the industry standard for affinity reagents for many years, but their use is often hampered by variable specificity and stability[10,11]. For binding helical peptides, the computationally designed helical scaffolds described in this paper have a number of structural and biochemical advantages. First, the extensive burial of the full length of an extended helix is difficult to accomplish with antibody loops[15], but very natural with matching extended α-helices

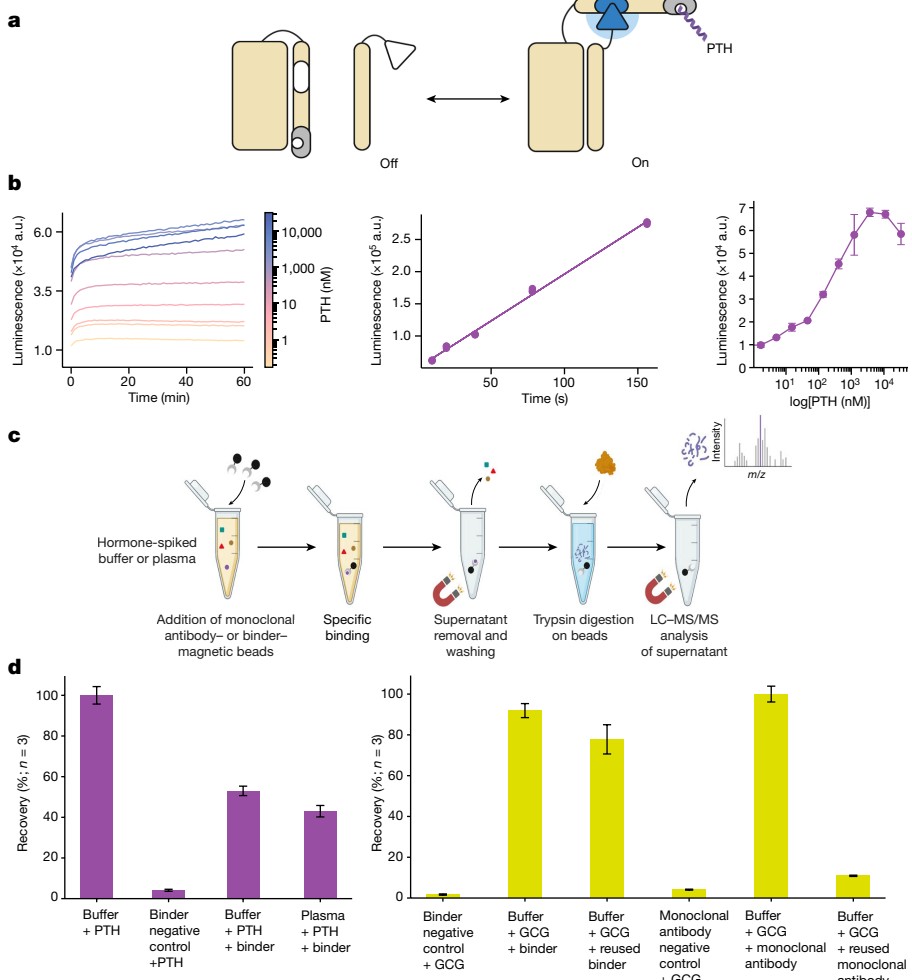

**Fig. 4 | Application of designed binders to sensing and detection. a**, The PTH lucCage biosensor. Cage and latch (left, beige), key (right, beige) and the PTH binder (grey) thermodynamically shift from the off to on state in the presence of PTH peptide target (purple). This conformational change brings two luciferase halves (inactive in white, active in blue) close together leading to luminescence. **b**, Left: titration of PTH results in luminescence increase ($n = 3$). Middle: response of lucCagePTH biosensor in the linear concentration range, indicating a 10 nM limit of detection (Supplementary Methods). Right: titration curve of 10 nM lucCagePTH + lucKey to various concentrations of PTH ($n = 3$). **c**, LC–MS/MS enrichment experiment schematic; the trypsin digestion step

was skipped for the GCG binder. **d**, Left: LC–MS/MS recovery percentages for triplicate measurements of an N-terminal tryptic peptide of PTH. The negative control comprised bovine serum albumin mixed with PTH in a buffer solution. Right: recovery percentage for triplicate measurements of intact GCG peptide normalized to the percentage recovery with a monoclonal antibody ($n = 3$). Following the first binding and elution experiments, beads were extensively washed and resuspended in PBS–CHAPS 0.1%, and then used in a second pulldown experiment. An unrelated binder attached to the magnetic beads mixed with GCG in buffer was used as a negative control. **a**,**c**, Created with BioRender.com.

in groove-shaped scaffolds. Second, designed scaffolds are more amenable to incorporation into sensors as illustrated by the lucCagePTH sensor. Third, they are more stable than antibodies, can be produced much less expensively, and can be easily incorporated into affinity matrices for enrichment of peptide hormones from human serum (the striking difference in the robustness of antibody-conjugated versus binder-conjugated beads to repeated use (Fig. 4d, right) highlights the differences in stability of the two modalities). Fourth, computational design avoids the need to immunize animals, which often mount weak responses to highly conserved bioactive molecules[42]. MS-based detection of peptides following enrichment using designed binders could provide a general route forwards for serological detection of a wide range of disease-associated peptide biomarkers.

Our results highlight the emergence of powerful new deep learning methods for protein design. The RF$_{joint}$ and RFdiffusion methods were both able to improve on initial Rosetta designs, and the Hallucination approach generated high-affinity binders without requiring pre-specification of the bound structures. Moreover, the RFdiffusion

method rapidly generated very tight (picomolar $K_d$ values) affinity and specific binders to several helical peptides. RFdiffusion was previously shown to be able to design binders to folded targets[8]; here we demonstrate further that it can be used to improve starting designs by partial noising and denoising, and can generate binders to peptides starting from no information other than the target sequence. To our knowledge, the Bim- and PTH-binding proteins diffused starting from random noise are the highest-affinity binders to any target (protein, peptide or small molecule) achieved directly by computational design with no experimental optimization. We expect both the RFdiffusion de novo peptide binder design capability and the ability to resample around initial designs (before or after experimental characterization) to be broadly applicable.

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

# Methods

## Computational methods

**Parametric design of groove-shaped scaffold library and use for binder design.** The parametric groove-shaped scaffold library was sampled using a random sampling approach, for which key parameters[22] were selected randomly from specific distributions. An even distribution of bundle 'lengths' was sampled, for which each parametric helix was 15–19 residues long. A supercoiling value was randomly selected from a biased distribution favouring more supercoiled scaffolds, given that these scaffolds were more likely to fail in the subsequent looping step (Fig. 1b and Supplementary Fig. 1). This biased sampling strategy was chosen to achieve a more uniform distribution of supercoiling within the final scaffold library, with sufficient numbers of highly supercoiled bundles. An average helix neighbour distance value was randomly selected from a Gaussian distribution informed by native protein helical bundle geometries (Fig. 1b and Supplementary Fig. 1). The distance of each helix from its neighbours was independently randomly selected from a much tighter Gaussian distribution centred at the preselected average helix neighbour distance value, to provide some noise within a given scaffold to helix distances and allow for heterogeneous amino acid selections (Supplementary Fig. 2). Values for helix phase and $z$ displacement were randomly sampled for each helix. The 'groove' consisting of three helices was first sampled as a helical bundle using the Crick parameterization of α-helical coiled coils, around an imaginary central helix where the target was to later be docked. Next, the two buttressing helices were sampled with the same parameterization, but moved radially outwards with randomly sampled helix neighbour distances as well as an additional randomly sampled tilt. This process was used to sample a set of 200,000 arrangements of 5 helices. Next, the Rosetta ConnectChainsMover[51] was used to loop this set into approximately 135,000 successful scaffold backbones. These backbones were designed and filtered using Rosetta[52] (including flexible backbone design) to yield a final library of 18,000 scaffolds. Backbones were filtered on metrics including buried nonpolar surface area per residue, Rosetta score per residue, percentage alanine, exposed hydrophobics per residue, and Rosetta 'holes'[53]. This library was used to design binders to different helical peptide targets using an adapted version of the miniprotein binder design computational pipeline used in ref. 7, in which only the binder interface was designed and the target was restricted to only rotamer repacking.

**RF$_{joint}$ Inpainting.** To sample around an initial putative binder, and to extend the binding interface to make additional contacts with the bound peptide, the RF$_{joint}$ Inpainting network was used[23], in conjunction with ProteinMPNN[24]. Rosetta-designed binders to PTH, GCG and NPY were used as input to RF$_{joint}$. RF$_{joint}$ is deterministic, and hence, to generate diversity, additional length was added (randomly and independently sampled) at the loop junctions between the binder helices. Additionally, one whole helix was completely rebuilt by RF$_{joint}$, to further permit diversification. RF$_{joint}$ designs were subsequently sequence redesigned with ProteinMPNN, validated and filtered in silico by AF2 with initial guess[6,28], and subsequently tested experimentally.

**Sequence threading to generate peptide binders.** We started from a library of several thousand all-helical scaffolds bound to designed single helices. We then threaded sequences of peptides of interest onto the bound single helix and filtered to obtain threaded conformations that maximized the number of target sequence positions that formed hydrophobic interactions at the interface to the binder scaffold[17,26]. The resulting binders were then redesigned in the presence of the threaded target sequence with ProteinMPNN[24] (forbidding cysteine) and the complex was predicted with AF2 with initial guess[6,28]. Another round of ProteinMPNN and AF2 + initial guess was carried out on the AF2 models that passed gate filters. Both rounds had gate filters of interface pAE < 10,

mean pLDDT > 92, predicted template modelling (pTM) score > 0.8 and RMSD to input backbone <1.75 Å. AF2 models from both rounds that passed gate filters were further filtered on AF2 metrics and filtered on Rosetta metrics to select sequences to order. Sequences were filtered against membrane insertion potential[54], contact_molecular_surface, ddG[7], interface pAE and monomer pAE[6].

**AF2 Hallucination for flexible peptide binder design.** Code for running Hallucination with AF2 was modified from ref. 31, with custom losses developed to promote binding of the Hallucinated protein to the input peptide sequence. AF2 model_4_ptm was used for all experiments. **Initial sequence sampling.** In line with ref. 31, the initial binder sequence was sampled randomly, with amino acid probabilities corresponding to background amino acid frequencies in BLOSUM62[55]. The target sequence (but no template structure) is also provided, separated by a chain break (+32 residue positional index offset). Residues were then mutated, with probabilities related to their background frequency in BLOSUM62. The number of amino acid changes proposed at each step is decayed throughout the trajectory (1,250 ×3 steps, 2,500 ×2 steps, 1,250 ×1 step). Multiple simultaneous amino acid changes initially helps speed up Hallucination, and a lower rate of changes later on allows more gradual refinement. To further speed up convergence, alterations were selectively made to residues with the lowest 50% of AF2 pLDDTs. **Losses used for Hallucination.** Losses used for Hallucination were as follows.

- pLDDT of the bound state: average pLDDT of the binder–peptide complex.
- pTM of the bound state: the pTM score of the binder–peptide complex.
- Radius of gyration: the radius of gyration was calculated as the mean squared distance of residues from the centre of mass of the protein. To approximately standardize the scaling with length of the protein, this was empirically normalized by dividing the radius of gyration by the radius of a sphere of volume related to the length of the Hallucinated protein.
- Contact probability: calculated as total probability that a residue in the target is in contact (closer than 8 Å) with the target peptide (the summed probability over the sub-8-Å bins of the distogram output from AF2). This was averaged across all target residues.
- Interface pAE: the mean pAE between the binder and peptide chains. For all examples shown in this work, the losses were weighted with relative weights of 1:1:0.1:3:5.

**Simulated annealing.** To optimize the designed binder, Monte Carlo simulated annealing was carried out, with a starting temperature of 0.01, and the half-life of the exponential decay set to 500 steps. Alterations were accepted or rejected using the Metropolis criterion. A total of 5,000 steps were carried out during design.

**ProteinMPNN.** Previous work has demonstrated that AF2 Hallucination yields adversarial sequences that do not work experimentally[31]. However, designs can be rescued with ProteinMPNN redesign of the sequences. Sixty-four sequences were designed per backbone, and were subsequently filtered on the basis of AF2 pLDDT, pTM, RMSD to the design model, RMSD of the monomer to the binder model (without the peptide), and Rosetta ddG. The precise values used for filtering were chosen to reduce the set down to 46 designs.

**Partial diffusion to optimize binders.** RFdiffusion was modified to allow the input structure to be noised only up to a user-specified time step instead of completing the full noising schedule. The starting point of the denoising trajectory is therefore not a random distribution. Rather, it contains information about the input distribution resulting in denoised structures that are structurally similar to the input (Fig. 2a). The AF2 models of the highest-affinity designs from Inpainting for GCG and NPY were used as inputs to partial diffusion. The models were subjected to 40 noising time steps out of a total of 200 time steps in the

noising schedule, and subsequently denoised. An auxiliary potential minimizing the radius of gyration of the binder–peptide complex was used (described below). Approximately 2,000 partially diffused designs were generated for each target. The backbones in the resulting library were sequence designed using ProteinMPNN (and ProteinMPNN after Rosetta FastRelax), followed by AF2 + initial guess[6]. The resulting libraries were filtered on AF2 pAE, pLDDT, RMSD to the design model, RMSD of the monomer to the binder model (without the peptide) and Rosetta ddG. The precise values used for filtering were chosen to reduce the set down to 96 designs for each target.

**De novo peptide binder design using RFdiffusion.** The AF2 model of the PTH peptide in the highest-affinity binder from Inpainting was used as input to RFdiffusion. For Bim, there was no previously designed binder and therefore the crystal structure of Bim[56] (PDB: 6X8O) was used as input. An auxiliary potential minimizing the radius of gyration of the binder–peptide complex was used during denoising (described below). Approximately 2,000 diffused designs were generated for each target. The backbones in the resulting library were sequence designed using ProteinMPNN (and ProteinMPNN following FastRelax), followed by AF2 + initial guess[6]. The resulting libraries were filtered on AF2 PAE, pLDDT, RMSD to the design model, RMSD of the monomer to the binder model (without the peptide) and Rosetta ddG. The precise values used for filtering were chosen to reduce the set down to 96 designs for each target.

**Radius of gyration potential.** RFdiffusion enables the use of external guiding potentials during inference, which help in the design of proteins with a certain desired property. The utility of these guiding potentials in designing symmetric oligomers and enzymes, as well as a description of how they are incorporated into the sampling procedure, is reported in ref. 8. In this work, we take advantage of guiding potentials to minimize the radius of gyration of the binder–peptide complex. The radius of gyration is calculated as the root mean square of the distance of all the Cα atoms from the centroid. It is more important to apply the potential at the initial denoising steps, and less so towards the end when the quaternary structure is largely fixed. Therefore, the scaling factor with which the gradients are multiplied has a cubic decay over the course of the denoising trajectory.

**Training RFdiffusion for designing binders to targets from sequence alone.** A modified version of RFdiffusion was trained to permit the design of protein binders to targets, for which only the sequence of the target was specified. The training strategy largely followed the training strategy used for the original RFdiffusion model, with some modifications. A summary is provided below.
**Overview of 'base' RFdiffusion training.** RFdiffusion[8] is a denoising diffusion probabilistic model fine-tuned from a pretrained structure prediction model; RoseTTAFold[57,58]. RFdiffusion is trained with a forward noising process that iteratively, over 200 time steps, noises residue translations and orientations to distributions that are indistinguishable from random distributions (three-dimensional Gaussian distribution and a uniform distribution on SO(3), respectively). RFdiffusion is then trained to reverse this corruption process, predicting the ground truth ($X_0$) at each time step of prediction. Mean-squared-error losses are used to minimize the error between the forward and reverse processes. Full training details are extensively described in ref. 8.
**Modifications to RFdiffusion for binder design to sequence inputs alone.** RFdiffusion was trained on both monomers (<384 amino acids) and heterocomplexes (one chain, denoted the 'binder chain', <250 amino acids) from the PDB. Coordinates were scaled by a factor of four, in line with the original RFdiffusion model. In 20% of cases, no sequence or structure was provided to the model (for unconditional generation). In the other 80% of cases, 20–100% of the protein was noised. In contrast to RFdiffusion, however, the structure of up to 50%

of the protein (monomer or 'target chain') was noised (diffused), while providing the sequence of those residues. Thus, RFdiffusion learns to condition its predictions on the sequence (without structure) of part of a protein (the monomer) or of a target to bind to. This version of RFdiffusion was trained for seven epochs.

**Computational filtering.** Precise metrics cutoffs changed for each design campaign to get to an orderable set, but largely focused on pAE (<10), pLDDT (>80) and Rosetta ddG (<−40)[6].

### Reporting summary

Further information on research design is available in the Nature Portfolio Reporting Summary linked to this article.

## Data availability

Atomic models of the GCG binders designed with Inpainting and partial diffusion (Fig. 2c), the Bim binder (Fig. 3d) and PTH peptide have been uploaded to the PDB with the accession codes 8GJG, 8GJI, 8T5E and 8T5F, respectively. Sequences of the binders described in this paper are in Extended Data Table 1.

## Code availability

Code for the parametric design pipeline can be found at https://github.com/proleu/peptide_paper/tree/main/projects/parametric_groove_design. Code to run RF$_{joint}$ Inpainting can be found at https://github.com/RosettaCommons/RFDesign. Computational notebooks for the sequence-threading pipeline can be found at https://github.com/proleu/peptide_paper/tree/main/projects/threading. Partial-diffusion code explanation and examples can be found at https://github.com/RosettaCommons/RFdiffusion#partial-diffusion. Code explanation and examples of binder design using RFdiffusion can be found at https://github.com/RosettaCommons/RFdiffusion#binder-design. An explanation of how to implement potentials, including the radius of gyration can be found at https://github.com/RosettaCommons/RFdiffusion#using-auxiliary-potentials. Code to run AF2 Hallucination for peptide design is available at https://github.com/RosettaCommons/AF2_peptide_hallucination.

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

**Acknowledgements** This work was supported with funds provided by a grant (U19 AG065156) from the National Institute for Aging (S.V.T., M.J.M., E.H., J.B., A.N.H., H.-H.H., I.D.L. and D.B.), a gift from Amgen (J.L.W.), the Audacious Project at the Institute for Protein Design (A.H.-W.Y. and D.B.), a gift from Microsoft Gift supporting Computational Protein Structure Prediction and Design at the Institute for Protein Design (D.J. and D.B.), the Washington State General Operating Fund supporting the Institute for Protein Design (P.V. and L.S.), a grant (INV-010680) from the Bill and Melinda Gates Foundation (D.J., J.L.W. and D.B.), an NIH NIBIB Pathway to Independence Award (A.H.-W.Y.; K99EB031913), a National Science Foundation Training Grant (EF-2021552; P.J.Y.L.), NERSC award BER-ERCAP0022018 (P.J.Y.L), the Open Philanthropy Project Improving Protein Design Fund (P.J.Y.L., G.R.L. and D.B.), The Donald and Jo Anne Petersen Endowment for Accelerating Advancements in Alzheimer's Disease Research (N.R.B.), an EMBO Postdoctoral Fellowship (grant number ALTF 292-2022; J.L.W.) and the Howard Hughes Medical Institute (D.B.). J.M.R. and F.H. were supported by the Novo Nordisk Foundation (NNF19OC0054441 to J.M.R.). H.-H.H. is supported by a postdoctoral fellowship provided by

the Partnership for Clean Competition. This work was additionally supported with funds provided by the grant T1D U01 DK121289 (J.B. and A.N.H.) and NSF Award 2021552 (J.N.). We thank Microsoft and AWS for gifts of cloud computing resources. Crystallographic diffraction data were collected at the Northeastern Collaborative Access Team beamlines at the Advanced Photon Source, which are funded by the National Institute of General Medical Sciences from the National Institutes of Health (P30 GM124165). This research used resources of the Advanced Photon Source, a US Department of Energy (DOE) Office of Science User Facility operated for the Department of Energy Office of Science by Argonne National Laboratory under contract number DE-AC02-06CH11357.

**Author contributions** D.B. directed the work. I.D.L. and S.V.T. designed, screened and experimentally characterized the parametrically designed groove scaffold peptide binders. P.J.Y.L. and S.V.T. designed, screened and experimentally characterized the threaded peptide binders. J.L.W. developed the Hallucination method for peptide binding. J.L.W., F.H. and J.M.R. designed and experimentally characterized the Hallucinated peptide binders. S.V.T. and J.L.W. designed and characterized the Inpainted binders. S.V.T. and P.V. designed, screened and experimentally characterized all the different classes of diffused peptide binders shown in this manuscript. S.R.G., A.M. and P.H. carried out additional scaled-up protein purification. P.J.Y.L., A.K.B., A.K. and J.D.L.C. obtained all the crystal structures shown in this manuscript. P.M.L., X.L. and M.L. synthesized fluorescently labelled peptides used during FP binding experiments.

J.L.W., D.J. and N.R.B. developed the RFdiffusion algorithm used for peptide binder design. H.-H.H., J.B., S.V.T, E.H., M.J.M. and A.N.H carried out the LC–MS/MS peptide detection. A.H.-W.Y. and S.V.T. designed and characterized the lucCagePTH biosensors and analysed the sensing experiments. J.N. provided research support and supervision. L.S. provided research strategy and funding acquisition support. M.E. and G.R.L. supported data analysis and yeast display binding screening. All authors reviewed and accepted the manuscript.

**Competing interests** D.B., S.V.T., P.J.Y.L., P.V., I.D.L., A.N.H., D.J., E.H., A.H.-W.Y., H.-H.H., J.L.W., M.J.M., N.R.B. and G.R.L. are inventors on a provisional patent application submitted by the University of Washington for the design and composition of the proteins created in this study.

**Additional information**
**Correspondence and requests for materials** should be addressed to Joseph L. Watson, Joseph M. Rogers or David Baker.

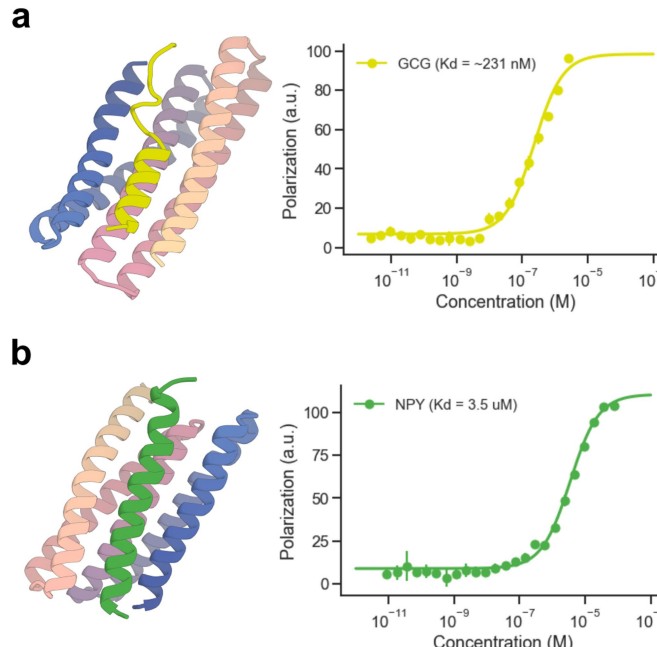

**Extended Data Fig. 1 | Low affinity RF$_{joint}$-Inpainted binders for NPY and GCG using extended parametric designs.** (**a**) Left: Design model (colour spectrum + yellow) of the tightest GCG binder. Right: FP titration (n = 4) for the tightest GCG binder indicates ~ 231 nM binding affinity (**b**) Left: Design model (colour spectrum + dark green) of the tightest NPY binder. Right: FP titration (n = 4) for the tightest NPY binder indicates 3.5 μM binding affinity.

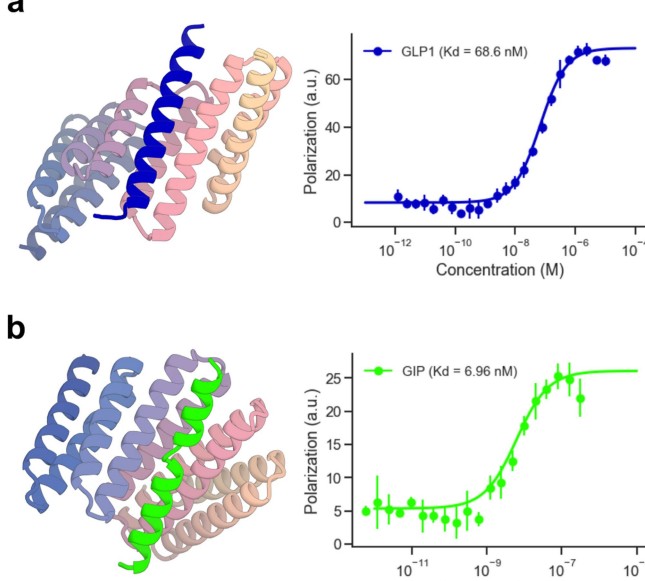

**a**

**b**

**Extended Data Fig. 2 | Additional binders made using threading and redesign.** (**a**) Left: Design model (colour spectrum + dark blue) of the tightest GLP1 binder. Right: FP titration (n = 4) for the tightest GLP1 binder indicates 68.8 nM binding affinity (**b**) Left: Design model (colour spectrum + green) of the tightest GIP binder. Right: FP titration (n = 4) for the tightest GIP binder indicates 6.96 nM binding affinity.

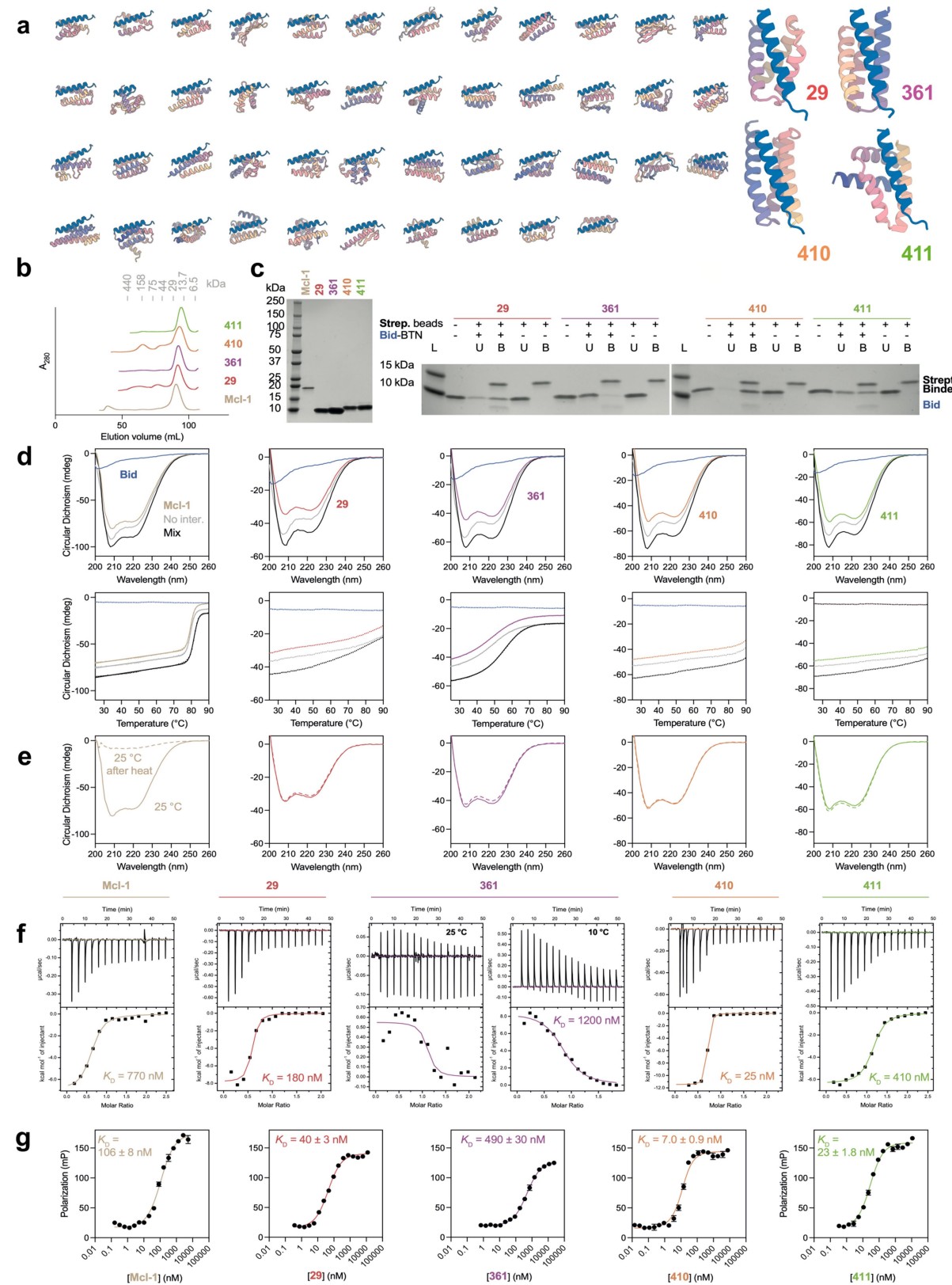

**Extended Data Fig. 3 |** See next page for caption.

**Extended Data Fig. 3 | Hallucinated Bid binders are stable and bind Bid peptide with high affinity.** (**a**) 46 Hallucinated designs tested for initial experimental screening. (**b**) 4 designs were chosen for expression without Bid peptide. All expressed as monomeric proteins (assessed by preparative SEC) and were pure by SDS-PAGE (n = 1). (**c**) All Hallucinations could be pulled-down by biotinylated Bid immobilised on streptavidin magnetic beads. B = bound to bead, U = unbound, in supernatant. L = ladder (n = 1). (**d**) Bid is unstructured in isolation by circular dichroism (CD), whereas all Hallucinations were helical in isolation, as predicted from the Hallucinated structure. A 1:1 molar ratio of binder:Bid (Mix) produced greater helical signal than that predicted by the isolated spectra (No inter.) suggesting binding is inducing helix formation (n = 1). (**e**) Melting with CD showed that Hallucinations were thermostable, and binding to Bid increased thermostability (where measurable) (n = 1). All Hallucinated binders would remain folded, or refold after heating and cooling, in contrast to the natural binder Mcl-1 which precipitated in the process. (**f**) ITC showed that Hallucinations bound to Bid, with µM to nM $K_d$s (n = 1). (**g**) FP measurements of designed Bid binders (n = 3).

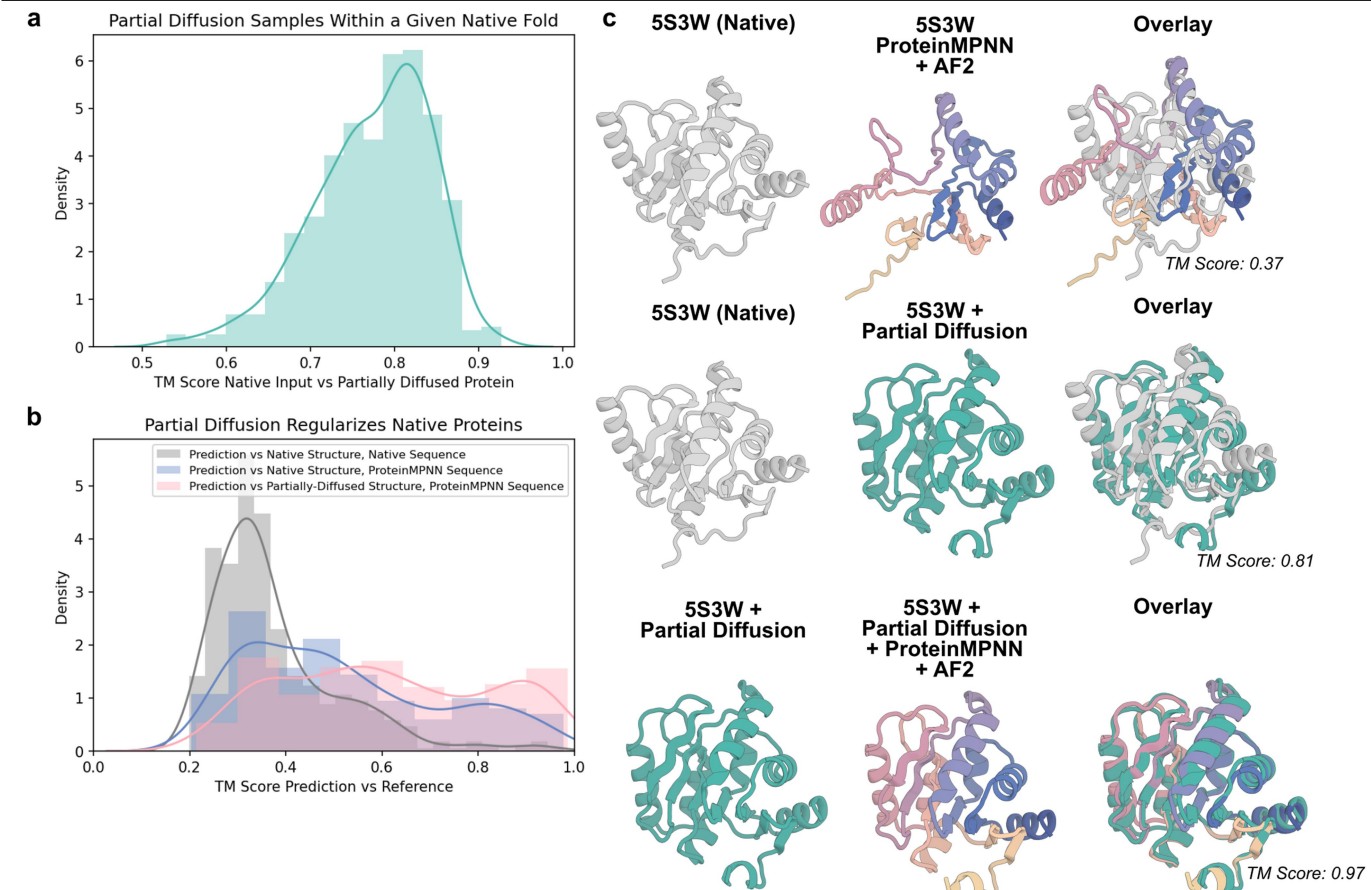

**Extended Data Fig. 4 | Partial diffusion increases designability of native proteins.** 500 native proteins of length 100 to 300 residues were selected from the PDB (< 3.5 Å resolution and no missing residues). Three different methods were applied to these proteins: 1) single sequence AlphaFold2 (AF2), 2) ProteinMPNN combined with AF2, and 3) partial diffusion (60 steps, noise = 1), ProteinMPNN and AF2. (**a**) Partial diffusion generates diverse protein conformations from the initial fold while maintaining the same overall fold, as indicated by the TM (Template Modeling) score exceeding 0.5. (**b**) The backbones resulting from partial diffusion exhibit higher designability compared to the native backbone, implying that they have been idealised for design purposes. (**c**) Visualisation of an example where partial diffusion + ProteinMPNN results in a significantly more designable protein relative to sequence redesign by ProteinMPNN on the native backbone.

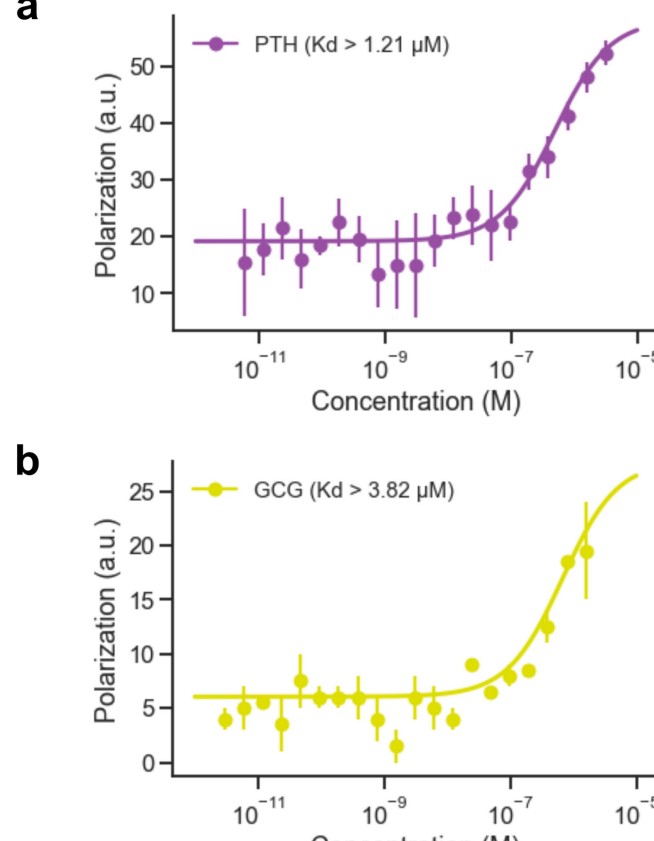

**Extended Data Fig. 5 | PTH and GCG binders designed with RFdiffusion.**
Representative binding data is shown for PTH (**a**) and GCG (**b**) binders designed
by providing sequence input alone. The binding affinities, as measured by FP
(n = 4), indicate low micromolar interactions with the respective peptide targets.

**a**

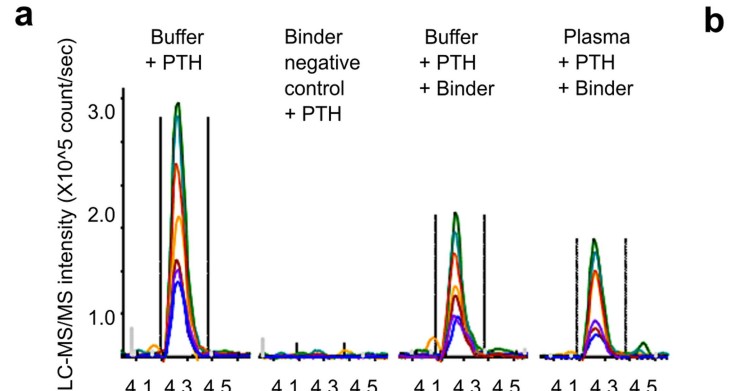
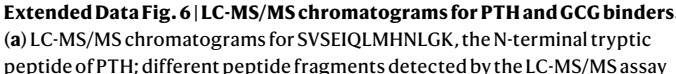

**b**

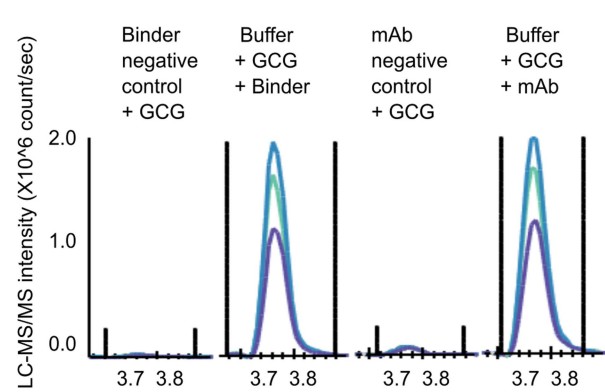

**Extended Data Fig. 6 | LC-MS/MS chromatograms for PTH and GCG binders.** (**a**) LC-MS/MS chromatograms for SVSEIQLMHNLGK, the N-terminal tryptic peptide of PTH; different peptide fragments detected by the LC-MS/MS assay are in different colours. (**b**) LC-MS/MS chromatograms for the intact GCG peptide HSQGTFTSDYSKYLDSRRAQDFVQWLMNT; different peptide fragments detected by the LC-MS/MS assay are in different colours.

# Extended Data Table 1 | Amino acid sequences of peptide binders

| Target | Computational method | Binder amino acid sequence |
|---|---|---|
| PTH | Parametric design | SQELIEELIKLAKELAEIKDEEERRKIKRELERLAEELKEAPASSLLRALALLVIALALIQAAESEEERERARELLERLE ELLRELEKQITDERFKEILRELEELAKELKKQL |
| PTH | Inpainting | SFELLEKLIELSKELYEVAKKYIETGDPELKKKLEEILKKIEEAYKELLESDAHPLEKALAKLILAEAYVVKSFAYISSG KDLEKAQEYLDKAKEILEEELLKLLEELKKEETDPEKLEIIEELEKIAEELLKEIEE |
| GCG | Inpainting | MLDELFSLLNKMFELSDKYRELRKELRKAIESGAPEEELRELLEKMLEIAKKLLELTKELKKLVEDVLKNNPDPVER AKAVLLYAVGVHILYSESSELEVIAERLGFKDIAEKAKEIADKARELKEEVKRKLREIREEVPDPEIRKAAEEAIEML ESNDKRLKEFRKL |
| NPY | Inpainting | GAAEKLAELYEKFKALREKALEVLKKAVEALENKADKETLLKLIKELKELAEKFEELAEEFERNAGESTTASLNATA AYMARIGLLAALLALAKAAGVPEEELEEIKRRIEETAKRAIEAAERLKALAEARGDTKHVAVGVEAVRMATELYELA QKIIDAF |
| SCT | Threading | SEELEERLREARERLEEARERLEEAREEGDLREMARALLEEARAVLEIARVAAEAGDDEALREAARRAGEVIRRA GEVGLRAAEEGDTETIREAMLAILEAQRASAVIALHLARDDPEVAEALRVIERLLRTAERALREGQLEVARLATEA VEALADAILRAREIGRPELVREAARLAEEARRLLEAALEALRAGDEEGARERLARARELIREIRERVRRA |
| GIP | Threading | SPKEKAERLIKEAKEAAEKAKEAAERSGLEEAKKAAEELTKLLEEAAARVAADPEDETKLRALEKIVEAAKEAVKA LEVAIESGDEQLIRAALSLVEAAVHLAKALLAKPESPLVDFGFELLKLAAKTLAAYAEGEDVDKIALKLKAISAMAEA LRLALAGDLERAARAAEEAVRYAIEAGDKELLRLAAEVAAYIARLAEEAGLEEVARRAREAAERAREAAK |
| GLP1 | Threading | SPEEEARRAAREAERAAREAREAARRLGDEESVRVAERLEREARRAERERDLELARRVLRAAEALRLALEGELL AREQGDELGVVVARMITLAARDSALGRGTPELARLLLRVARALLEGDLEEVVRSLAEIAKREIGTERALLAVEAIKL VALESIEEGDFETAELAIEKLREIAEEFEGTEVAEKAREAIEEIEKKKREAE |
| Bid | Hallucination | TPEDYRRAAELIKEIAREAERYAEGEISAEEALARIRRLRAELEECYEHGLDAVGRSYVDQARPLIDEIERLLQEKL DAE |
| GCG | RF Diffusion (partial) | SMEKLAEIMQEIIEAYQEVKDAFFKFIKAVHEGAPEEELKKYLEKMKEALEKMKELLERLEKEAKKVIEENKDKKLE LKVLLMLRLAYLLLKVSIELTKIAAEKLGDKELVEELEKESKEVEKKIKELEERIKKLLEEVDDEELKEAYKEVEEME KEAEKFLEKMRKV |
| NPY | RF Diffusion (partial) | GMEERRKELLEKLKKLKEEVVELFRELAQALRDGASKERLEEIRERAEKLAEEAKKVAEELEKLAEGDAVLQLYLA EAYALEAAALTIEAVAAAELGASKEELEKIKEKIEEALKKAEEAMKKALAEAKARGRERLVRLIEEARKEFEKLSKAI KELLEQV |
| PTH | RF Diffusion | MREKLEEMLEEFNEVIDELIEITKEDAPELEELRERAEEAVENERLDELEEILDELEVIILEAMFRDLSAAIEMTKAKN DKEKLKELLKQLEELEKRIKELLERAKKRGNKKIIEKLEKLLKEVEKLKKEIEEYLK |
| Bim | RF Diffusion | EEERKEKREKVRAGLKRAIAELPAEVAARCLALLDDASDEEFIEAVLEVLEAMREALVAMAREGRLDAVRRATSHI NEVLVDAAELALEKGREYFRRLCLIVCDMMIELIRLEPEQTPELRRIRERLEEIRRRLE |
| PYY | RF Diffusion (sequence only) | GLEEAEKLLEEIFANFEEIVELIKKNIGTERGKKLLKVFVATVDLILARLEQGADLAELAELVKEIAELAKDEEGLEEA EKLVKELTAAR |

**Extended Data Table 2 | Crystallographic data collection and refinement**

| | GCG_partdiff (8GJI) | GCG_inpaint (8GJG) | Bim_fulldiff (8T5E) | PTH (8T5F) |
|---|---|---|---|---|
| **Data collection** | | | | |
| Space group | $P\,2_1\,2_1\,2_1$ | $P\,2_1$ | $P\,2_1\,2_1\,2_1$ | $P\,4\,2_1\,2$ |
| Cell dimensions | | | | |
| $a$, $b$, $c$ (Å) | 31.26, 50.92, 91.92 | 37.79, 45.90, 63.60 | 25.79, 66.67, 74.30 | 91.32, 91.32, 37.73 |
| $\alpha$, $\beta$, $\gamma$ (°) | 90, 90, 90 | 90, 97.31, 90 | 90, 90, 90 | 90, 90, 90 |
| Resolution (Å) | 91.93 - 1.81 (1.88 - 1.81) | 63.09 - 1.95 (2.00 - 1.95) | 74.30 - 3.00 (3.18 - 3.00) | 40.84 - 1.99 (2.04 - 1.99) |
| $R_{merge}$ | 0.099 (1.581) | 0.073 (2.001) | 0.064 (0.173) | 0.120 (2.069) |
| $I$ / $cI$ | 8.3 (0.90) | 10.10 (0.60) | 17.8 (8.8) | 21.8 (1.3) |
| Completeness (%) | 99.60 (99.40) | 98.50 (96.30) | 99.9 (100) | 98.6 (94.6) |
| Redundancy | 6.2 (6.0) | 6.7 (6.8) | 7.2 (7.9) | 15.1 (15.2) |
| | | | | |
| **Refinement** | | | | |
| Resolution (Å) | 45.96 - 1.81 (1.88 - 1.81) | 63.09 - 1.95 (2.00 - 1.95) | 49.62 - 3.00 (49.62 - 3.00) | 40.84 - 1.99 (2.19 - 1.99) |
| No. reflections | 13875 (1327) | 15425 (2387) | 2861 (445) | 11302 (738) |
| $R_{work}$ / $R_{free}$ | 0.2080 (0.3752)/ 0.2552 (0.4485) | 0.2087 (0.4205)/ 0.2488 (0.4445) | 0.2398 (0.2398)/ 0.2617 (0.2617) | 0.2201 (0.2506)/ 0.2494 (0.3372) |
| No. atoms | | | | |
| Protein | 1579 | 1539 | 1244 | 853 |
| Ligand/ion | 0 | 0 | 0 | 0 |
| Water | 24 | 26 | 0 | 26 |
| $B$-factors | | | | |
| Protein | 45.14 | 68.55 | 77.56 | 61.14 |
| Ligand/ion | 0 | 0 | 0 | 0 |
| Water | 47.64 | 69.57 | n/a | 62.39 |
| R.m.s. deviations | | | | |
| Bond lengths (Å) | 0.012 | 0.002 | 0.003 | 0.010 |
| Bond angles (°) | 1.12 | 0.440 | 0.500 | 1.04 |

*Single Crystal used for each data collection. *Values in parentheses are for highest-resolution shell.

2022-12-19949B

# Reporting Summary

## Statistics

For all statistical analyses, confirm that the following items are present in the figure legend, table legend, main text, or Methods section.

| n/a | Confirmed | |
|---|---|---|
| ☐ | ☒ | The exact sample size (*n*) for each experimental group/condition, given as a discrete number and unit of measurement |
| ☐ | ☒ | A statement on whether measurements were taken from distinct samples or whether the same sample was measured repeatedly |
| ☐ | ☒ | The statistical test(s) used AND whether they are one- or two-sided *Only common tests should be described solely by name; describe more complex techniques in the Methods section.* |
| ☒ | ☐ | A description of all covariates tested |
| ☒ | ☐ | A description of any assumptions or corrections, such as tests of normality and adjustment for multiple comparisons |
| ☐ | ☒ | A full description of the statistical parameters including central tendency (e.g. means) or other basic estimates (e.g. regression coefficient) AND variation (e.g. standard deviation) or associated estimates of uncertainty (e.g. confidence intervals) |
| ☒ | ☐ | For null hypothesis testing, the test statistic (e.g. *F*, *t*, *r*) with confidence intervals, effect sizes, degrees of freedom and *P* value noted *Give P values as exact values whenever suitable.* |
| ☒ | ☐ | For Bayesian analysis, information on the choice of priors and Markov chain Monte Carlo settings |
| ☒ | ☐ | For hierarchical and complex designs, identification of the appropriate level for tests and full reporting of outcomes |
| ☒ | ☐ | Estimates of effect sizes (e.g. Cohen's *d*, Pearson's *r*), indicating how they were calculated |

*Our web collection on statistics for biologists contains articles on many of the points above.*

## Software and code

Policy information about availability of computer code

| Data collection | Parametric groove design, Threading, RFjoint, Hallucination, RFdifussion, ProteinMPNN, AlphaFold2 |
|---|---|
| Data analysis | Matplotlb 3.5.3, SciPy 1.7.3, Seaborn 0.11.2, PyMOL 2.5.0 |

For manuscripts utilizing custom algorithms or software that are central to the research but not yet described in published literature, software must be made available to editors and reviewers. We strongly encourage code deposition in a community repository (e.g. GitHub). See the Nature Portfolio guidelines for submitting code & software for further information.

## Data

Policy information about availability of data

All manuscripts must include a data availability statement. This statement should provide the following information, where applicable:
- Accession codes, unique identifiers, or web links for publicly available datasets
- A description of any restrictions on data availability
- For clinical datasets or third party data, please ensure that the statement adheres to our policy

Three protein-peptide crystallographic structures obtained for glucagon and BIM peptide have been made available.

# Research involving human participants, their data, or biological material

Policy information about studies with [human participants or human data](). See also policy information about [sex, gender (identity/presentation), and sexual orientation]() and [race, ethnicity and racism]().

| | |
|---|---|
| Reporting on sex and gender | *Use the terms sex (biological attribute) and gender (shaped by social and cultural circumstances) carefully in order to avoid confusing both terms. Indicate if findings apply to only one sex or gender; describe whether sex and gender were considered in study design; whether sex and/or gender was determined based on self-reporting or assigned and methods used.*<br>*Provide in the source data disaggregated sex and gender data, where this information has been collected, and if consent has been obtained for sharing of individual-level data; provide overall numbers in this Reporting Summary. Please state if this information has not been collected.*<br>*Report sex- and gender-based analyses where performed, justify reasons for lack of sex- and gender-based analysis.* |
| Reporting on race, ethnicity, or other socially relevant groupings | *Please specify the socially constructed or socially relevant categorization variable(s) used in your manuscript and explain why they were used. Please note that such variables should not be used as proxies for other socially constructed/relevant variables (for example, race or ethnicity should not be used as a proxy for socioeconomic status).*<br>*Provide clear definitions of the relevant terms used, how they were provided (by the participants/respondents, the researchers, or third parties), and the method(s) used to classify people into the different categories (e.g. self-report, census or administrative data, social media data, etc.)*<br>*Please provide details about how you controlled for confounding variables in your analyses.* |
| Population characteristics | *Describe the covariate-relevant population characteristics of the human research participants (e.g. age, genotypic information, past and current diagnosis and treatment categories). If you filled out the behavioural & social sciences study design questions and have nothing to add here, write "See above."* |
| Recruitment | *Describe how participants were recruited. Outline any potential self-selection bias or other biases that may be present and how these are likely to impact results.* |
| Ethics oversight | *Identify the organization(s) that approved the study protocol.* |

Note that full information on the approval of the study protocol must also be provided in the manuscript.

# Field-specific reporting

Please select the one below that is the best fit for your research. If you are not sure, read the appropriate sections before making your selection.

☒ Life sciences ☐ Behavioural & social sciences ☐ Ecological, evolutionary & environmental sciences

For a reference copy of the document with all sections, see [nature.com/documents/nr-reporting-summary-flat.pdf]()

# Life sciences study design

All studies must disclose on these points even when the disclosure is negative.

| | |
|---|---|
| Sample size | Variable depending on analysis performed. Detailed in figure legends. |
| Data exclusions | None |
| Replication | Each data set contains many (n reported in figure legends) independent measurements |
| Randomization | N/A |
| Blinding | N/A |

# Reporting for specific materials, systems and methods

We require information from authors about some types of materials, experimental systems and methods used in many studies. Here, indicate whether each material, system or method listed is relevant to your study. If you are not sure if a list item applies to your research, read the appropriate section before selecting a response.

## Materials & experimental systems

| n/a | Involved in the study |
|-----|----------------------|
| ☒ | ☐ Antibodies |
| ☒ | ☐ Eukaryotic cell lines |
| ☒ | ☐ Palaeontology and archaeology |
| ☒ | ☐ Animals and other organisms |
| ☒ | ☐ Clinical data |
| ☒ | ☐ Dual use research of concern |
| ☒ | ☐ Plants |

## Methods

| n/a | Involved in the study |
|-----|----------------------|
| ☒ | ☐ ChIP-seq |
| ☒ | ☐ Flow cytometry |
| ☒ | ☐ MRI-based neuroimaging |

