## [Peer Review File · Nature]

Manuscript Title: De novo design of high-affinity binders of bioactive helical peptides

Redactions – unpublished data

Reviewer Comments & Author Rebuttals

Reviewer Reports on the Initial Version:

Referees' comments:

Referee #1:

In the paper, “De Novo design of high-affinity protein binders to bioactive helical peptides”, Vasquez Torres et al. describe a test of three different design strategies for binders in helical peptides. In short, they demonstrate that RFdiffusion is superior to the other methods.

This paper is strongly related to the accompanying paper by Watson et al., and it could be incorporated into that paper if space allowed. However, I assume that both the authors and the publisher might see an advantage in publishing two papers, particularly as the first authors of the papers are different. But this is an editorial decision.

Major

From a computational point of view, this paper does not add much to the Watson paper. However, the use of denoising from a starting structure is exciting. It would be interesting if this also could be applied to the manual design and to examine how the structural models (and designed sequences) change by these (assumed) small structural changes.

Minor

I find the structural differences between the differently designed models surprisingly small (Fig. S5). Although there is an indication that the RFdiffusion models are “better”, there is a large overlap between the two sets of models, i.e. it is not very clear what the difference is. Could the authors try to expand the analysis to provide a better explanation?

Referee #2:

Vasquez Torres et al use a variety of computational approaches to design small helical proteins that bind to helical peptides. Each of the computational approaches produced designs with some binding affinity to their targets. The authors then built a luminescent sensor for the parathyroid hormone peptide from one of the binder designs, based on a construct from previously published work. In general, this paper shows good design success for the desired task (binding helical peptides), but it seems incomplete; the authors are missing descriptions of key methods and do not fully characterize their designed proteins. The authors provide no scripts, no design models, no sequences, and very little information concerning their protocols and

filtering procedures. This significantly impacts reproducibility of the work. I hope that the authors can address this as well as additional questions/comments below.

The authors have not sufficiently established the scope of the design challenge. Why is this problem hard? It seems the authors struggle with defining why designing proteins to bind to helical peptides should be difficult, saying, e.g., “[peptide backbone] interactions cannot be made with alpha helical peptides due to the extensive internal backbone - backbone hydrogen bonding”. Why would establishing peptide-backbone interactions be critical for this task? Some two-helix coiled coils associate with pM to fM affinity using sidechain packing alone. Indeed, the local H-bonding of a helix might be seen as an advantage in protein design, as the self-contained nature of the backbone is such that only sidechains need to be considered during design (for which packing interactions and the structure/sequence relationship have been studied for many decades to great success); so I am struggling to understand how the helical nature of the peptide is being defined as a hurdle. The authors should elaborate.

Is the design of proteins that bind to helical peptides really an outstanding challenge? DeGrado and coworkers showed the successful computational design of high affinity binders (5 nM Kd) to a helical calmodulin-binding peptide in 1998, which the authors do not cite in their manuscript. See G. Ghirlanda, J. D. Lear, A. Lombardi, W. F. DeGrado, From synthetic coiled coils to functional proteins: automated design of a receptor for the calmodulin-binding domain of calcineurin. *J. Mol. Biol.* 281, 379–391 (1998). In the submitted manuscript, Vazquez Torres et al show that they can achieve mid to low nM (and in two cases sub-nM, although stoichiometry is not reported) affinity binders to helical peptides using a variety of computational approaches, which is very nice to see, but the abundance of successful designs belies the challenging nature of the problem domain. My enthusiasm for this work would grow immensely if the authors could show successful design of arbitrary peptides, especially those with no secondary structure.

Of the design methods mentioned in this work, hallucination seems the most promising and general, since it does not need a predefined structure of the target peptide. However, the authors did not demonstrate that the approach is plausible for more difficult targets where the structure of the target peptide is not helical or unknown. Nor did they adequately describe the procedure for computing hallucinated binders.

It is not clear if each design approach (parametric, parametric+inpainting, hallucination, sequence-threading, diffusion) was set against all 6 peptides targeted in this work, or if some

approaches targeted only a subset of peptides. For example, did the parametric design approach fail to find any binders to SCT, Bid, or Bim, or where these peptides not targeted with this approach?

Fig 1b and c are vague and could be improved if the actual bundle topology that is being explored is depicted. The cartoon is too imprecise and potentially misleading.

Please show the data for the “pilot experiments” mentioned in the supplement (Identification of weak binder hits from parametric designs in pilot experiment). I am not sure why these are called “pilot experiments” instead of “experiments”. What are the additional experiments that indicated > 100 nM binding affinities?

Parametric design of groove-shaped scaffold library and use for binder design. “A supercoiling value was randomly selected from a biased distribution favoring more supercoiled scaffolds, given these scaffolds were more likely to fail in the subsequent looping step.” Please explain. This section would greatly benefit from a figure describing what the parameters and distributions are that are being sampled. The “adapted version of the miniprotein binder design pipeline” needs to be clearly described. The computational method as written in the current manuscript is not reproducible. How were the initial parametric binders designed? Was any backbone flexibility allowed during the design with peptide?

Design of BIM peptide binders. This section is inadequately described. “External potentials were used to promote interactions between the binder and target - specifically, the radius of gyration of the complex was minimized.” How? The authors need to significantly elaborate on the exposition of the methods.

It is now common practice to deposit plasmids to addgene for distribution and reproducibility. I do not think the Baker lab has deposited the petcon3 plasmid, which they commonly use for yeast display. I urge the authors to deposit the plasmid to addgene or publish the plasmid map.

“There is considerable interest in their sensitive and specific quantification, which currently relies on antibodies that require substantial resources to generate, can be difficult to produce with high affinity, and often have less-than-desirable stability and reproducibility⁵.” Could the authors cite some papers other than from the Baker lab to qualify the “considerable interest” mentioned here?

Figure 2c. The hallucinated protein trajectory does not appear to be for the same protein shown to the left of the plot, which appears to be all helical. The trajectory has beta sheet. The binding isotherm only has one datapoint in the transition from fully unbound to fully bound. As such, the dissociation constant is sensitive to small errors in the single point that describes the transition. It would be better to measure more data points around this region in order to increase the confidence in the reported K_d value.

Please describe the yeast display protocol more thoroughly. Were the eblocks transformed into different cultures, were they pooled, etc? Were full eblocks ordered for each of the 192 designs?

The authors screened 12 designs for each target to evaluate the parametric design approach. They then screen 192 designs for a single target (PTH) using additional inpainting. They found a tighter binder. Does this necessarily mean that inpainting is able to produce better binders? I agree that additional buried surface area should lead to higher affinity, but there is likely a correlation between finding a tighter binder and testing more sequences per target. How much does inpainting matter here? Could the authors have just created longer parametrically generated bundles and/or simply screened more sequences experimentally?

“and the highest-affinity binder had 19% greater surface area contacting the target peptide.” There are no experiments that support this statement. I believe the authors mean that the computational model of the highest-affinity binder had 19% greater surface area contacting the target peptide.

“We started from a library of scaffolds that contained single helices bound by pseudorepetitive helical scaffolds.” Where did this library come from? More detail is required, as this is otherwise not reproducible.

“The binders were then redesigned in the presence of the threaded target sequence with ProteinMPNN and the complex was predicted with AF2 (with initial guess) and filtered on AF2 and Rosetta metrics.” Describe initial guess in the supplement, as well as each of the metrics used for filtering.

“Following size exclusion chromatography (SEC) purification of the monomer fraction...” Where is the SEC data? How much of these designs is not monomer? Does the collected monomer fraction re-equilibrate to higher order oligomerization state? This is important since this effects stoichiometry of binding and therefore could impact the reported binding affinity.

Fig S4b, the chromatogram traces appear to be cut off prematurely. Please indicate which elution time corresponds to the expected MW.

“The average contact molecular surface for the partially diffused GCG binders and NPY increased by 33% and 29% respectively compared to the starting models, and the Rosetta ddG improved by 29% and 21% (Fig. S5a, S5b).” What is this metric for the designs that experimentally bound their target? For the highest affinity binders?

The authors note that the “human”- vs “machine”- based solutions to this design problem were largely similar. Perhaps this similarity is not so surprising, given that there are many examples of helical bundle proteins in the PDB. Designers are quite familiar with helical bundles, as are the “machines.” RFdiffusion is highly biased to generate small helical proteins. The application mentioned here would be far more impressive to achieve on peptides with little secondary

structure, but I imagine that RFdiffusion and other deep-learning methods will struggle with this task due to the much sparser training data. The authors should discuss these limitations or consider showing that the approach could be feasible for arbitrary peptides.

What do the authors mean by “wrong positioning of the antigen-binding site during sensor immobilization” and how does their approach address this?

There are no structures of apo or holo proteins. How do the authors know the designs are atomically accurate? Were any mutational studies performed to determine effects on binding affinity?

Fig 5a. This schema is confusing and mislabeled in the figure legend (There is no red color in the image.). Please update to clearly show a labeled construct. Where is the peptide, where is the designed binder?

Fig 5b, shouldn't the emission saturate at some concentration? Is the affinity of the cage so high for the fused binder protein that the peptide cannot effectively outcompete it despite its pM affinity? Perhaps the authors could either run this experiment to saturation or provide an analysis for why the fluorescence does not saturate at 10 microM concentration of peptide.

Fig S3, what is contact probability? Where are the scripts that ran these design trajectories?

“The plasma samples used were de-identified leftover clinical samples obtained from the clinical laboratories at the University of Washington Medical Center.” What is a “leftover” clinical sample?

Please indicate why PTH needed to be supplemented in the clinical plasma samples. Is it because the physiological levels (pg/ml) are too low for measurable enrichment by the peptide binder?

Please comment on if the majority of these designs needed to be co-expressed with the target peptide to prevent aggregation and if this presents a challenge to using these for clinical diagnostic purposes. Only the RFdiffusion-derived proteins were commented to express without need for co-expression of target peptide.

Fig 4c. What is the stoichiometry of binding? The data looks like it might deviate from the expected 1:1 stoichiometry. This might affect the value of the dissociation constant.

Where are the SEC chromatograms of the RFdiffusion-derived binders showing that they are monomeric?

Referee #3:

This is a companion paper to the RFDiffusion paper, as such my concerns about reproducibility of the model used in the aforementioned paper also apply to this work.

This work is concerned with designing binders to small helical peptides. As the authors point out, this is challenging for traditional methods as the binding interfaces for peptides are smaller than in design of binders for other targets, also peptides are often unstructured in isolation. This makes this problem well suited for machine learning methods, which have proven much more successful at predicting protein complex structures in cases where the proteins are only structured upon binding.

From my perspective this is a solid application work following what is a fairly well trodden path by now: take the latest structure generation model, redesign the sequence with ProteinMPNN and then filter out structures based on AlphaFold2 predictions. I am not surprised that this works well given the many similar successes of this approach in the past; the novelty here is using RFDiffusion as the generative model.

The experimental evaluation here looks very solid to me. Overall to me this looks like a good application paper; however I am not an expert on the biology part and as such can not really comment on it.

Alexander Pritzel

Author Rebuttals to Initial Comments:

Referee #1:

In the paper, “De Novo design of high-affinity protein binders to bioactive helical peptides”, Vazquez Torres et al. describe a test of three different design strategies for binders in helical peptides. In short, they demonstrate that RF Diffusion is superior to the other methods.

This paper is strongly related to the accompanying paper by Watson et al., and it could be incorporated into that paper if space allowed. However, I assume that both the authors and the publisher might see an advantage in publishing two papers, particularly as the first authors of the papers are different. But this is an editorial decision.

Major

From a computational point of view, this paper does not add much to the Watson paper. However, the use of denoising from a starting structure is exciting. It would be interesting if this also could be applied to the manual design and to examine how the structural models (and designed sequences) change by these (assumed) small structural changes.

We agree with the reviewer, and have incorporated detailed analysis of the structural changes brought about by the partial diffusion method. First, we have carried out partial diffusion on the classic colicin-immunity protein system, starting with pairs that do not interact experimentally. We find that partial diffusion introduces small backbone shifts that optimize interactions with the target, in some cases recapitulating structural changes observed within the colicin-immunity protein family. We have incorporated this into the manuscript as Supplementary Fig. S11. Second, we have solved crystal structures of designs before and after partial diffusion for glucagon binders, which allows precise pinpointing of the subtle structural shifts which bring about the large increase in affinity. We highlight these results in the new Fig. 2c-e. Third, we have explored the broader applicability of partial diffusion by using it to refine a large set of native backbones; we find that this systematically increases designability (Supplementary Fig. S10).

Minor

I find the structural differences between the differently designed models surprisingly small (Fig. S5). Although there is an indication that the RF Diffusion models are “better”, there is a large overlap between the two sets of models, i.e. it is not very clear what the difference is. Could the authors try to expand the analysis to provide a better explanation?

While the topologies of the designs made using hallucination, inpainting and RF Diffusion have the same overall groove shape, the details of the interactions with the peptides are quite different. This is evident both in overall statistics, as the average contact molecular surface for the partially diffused GCG binders and NPY increased by 33% and 29% respectively compared to the starting models, and the Rosetta ddG improved by 29% and 21% and most clearly, in the

comparison of the crystal structures of the glucagon binding designs pre and post diffusion mentioned above. We have clarified this in the revised manuscript.

Referee #2:

Vazquez Torres et al use a variety of computational approaches to design small helical proteins that bind to helical peptides. Each of the computational approaches produced designs with some binding affinity to their targets. The authors then built a luminescent sensor for the parathyroid hormone peptide from one of the binder designs, based on a construct from previously published work. In general, this paper shows good design success for the desired task (binding helical peptides), but it seems incomplete; the authors are missing descriptions of key methods and do not fully characterize their designed proteins.

We have sought to address this concern by providing detailed descriptions of the key computational methods in the revised manuscripts along with more detailed characterization of the designed proteins. The highlight of the latter are three new crystal structures showing the detailed atomic interactions, which are very close to those in the design models.

The authors provide no scripts, no design models, no sequences, and very little information concerning their protocols and filtering procedures.

All current scripts, design models and sequences are now provided in the revised manuscript, along with a more detailed description of the methods (Supplementary Computational Methods, Supplementary Table 2). To promote use of the newer RF Diffusion method, which is 3-4 orders of magnitude faster than the Hallucination method, legacy versions of the Hallucination code will be made available at publication, for reproducibility.

This significantly impacts reproducibility of the work. I hope that the authors can address this as well as additional questions/comments below.

We hope that the additions to the manuscript fully resolve the reviewers concerns.

The authors have not sufficiently established the scope of the design challenge. Why is this problem hard? It seems the authors struggle with defining why designing proteins to bind to helical peptides should be difficult, saying, e.g., “[peptide backbone] interactions cannot be made with alpha helical peptides due to the extensive internal backbone - backbone hydrogen bonding”. Why would establishing peptide-backbone interactions be critical for this task? Some two-helix coiled coils associate with pM to fM affinity using sidechain packing alone. Indeed, the local H-bonding of a helix might be seen as an advantage in protein design, as the self contained nature of the backbone is such that only side chains need to be considered during design (for which packing interactions and the structure/sequence relationship have been studied for many decades to great success); so I am struggling to understand how the helical nature of the peptide is being defined as a hurdle. The authors should elaborate.

We thank the reviewer for suggesting that we make this point clearer. While it is certainly the case that two helix coiled coils associate with very high affinity, it is challenging to design coil coil sequences that will bind with high specificity to a specific target sequence, and the designs are almost never monomeric (since amphipathic helices will almost always either self associate or associate with other monomers). Also, coiled coil based binding strategies cannot readily deal with the non-helical portions of many of the targets described here. We make these points clearer in the revised manuscript.

Is the design of proteins that bind to helical peptides really an outstanding challenge? DeGrado and coworkers showed the successful computational design of high affinity binders (5 nM Kd) to a helical calmodulin-binding peptide in 1998, which the authors do not cite in their manuscript. See G. Ghirlanda, J. D. Lear, A. Lombardi, W. F. DeGrado, From synthetic coiled coils to functional proteins: automated design of a receptor for the calmodulin-binding domain of calcineurin. *J. Mol. Biol.* 281, 379–391 (1998).

We thank the reviewer for pointing out this paper which has now been cited in the manuscript. The DeGrado binders highlighted by the reviewer are two interacting parallel helical peptides connected by a disulphide (which then interact with the target calcineurin peptide). With this approach its easy to imagine domain swapping to form oligomers, and indeed, the designs form dimers. These dimers need to separate before binding the target peptide. The quoted 7 nM Kd is only after correcting for this effect, the actual measured affinity was in the μ M range. In a similar DeGrado paper (<https://pubs.acs.org/doi/10.1021/ja972973d>), the binder has two antiparallel helices, one peptide chain. But, only μ M Kd. Again, domain-swapping is very easy to imagine, and they do find that the binder forms a dimer. Like the other paper, this dimer needs to separate before binding the target peptide. Lastly, the DeGrado approach could be argued to lack generality, requiring a target helix of a particular length and sequence composition. In contrast, our approach(es) can target more diverse helical peptides and, by producing monomeric designs of high stability, avoid the downsides of DeGrado-style small coiled-coils, namely that they are oligomerization-prone and can lose effective affinity to their target as a result. Finally, we could argue that the apparent lack of progress since 1997 speaks to the fact that this is not a trivial design problem.

In the submitted manuscript, Vazquez Torres et al show that they can achieve mid to low nM (and in two cases sub-nM, although stoichiometry is not reported) affinity binders to helical peptides using a variety of computational approaches, which is very nice to see, but the abundance of successful designs belies the challenging nature of the problem domain. My enthusiasm for this work would grow immensely if the authors could show successful design of arbitrary peptides, especially those with no secondary structure.

In the revised manuscript we describe binders to peptides with a mixture of helical and non-helical regions. We extend RF Diffusion to enable design starting from target sequence alone so both the backbone of the non-helical portion of the peptide and the backbone of the binder can adapt to one another during the diffusion process (Fig. 3e).

Beyond the scope of this paper, we have used the free diffusion given only the sequence of the target peptide method introduced in this paper to design binders to a variety of peptides in different conformations, as illustrated in the attached slide deck and Appendix at the end of this document.

Of the design methods mentioned in this work, hallucination seems the most promising and general, since it does not need a predefined structure of the target peptide. However, the authors did not demonstrate that the approach is plausible for more difficult targets where the structure of the target peptide is not helical or unknown. Nor did they adequately describe the procedure for computing hallucinated binders.

We thank the reviewer for pointing out this advantage of Hallucination in allowing the backbone to be flexible. Given the power and massively improved computational efficiency of RF Diffusion (described both here and in our accompanying manuscript, Watson *et al.*), we have now trained an RFDiffusion model capable of binding to sequences *without* a predefined structure (as in Hallucination). We now add *in silico* and experimental data demonstrating that this model is highly effective at generating *in silico*-validated binders (Fig. 3e). We fully describe the training and benchmarking of this fine-tuned RFDiffusion model in the Methods section.

It is not clear if each design approach (parametric, parametric+inpainting, hallucination, sequence-threading, diffusion) was set against all 6 peptides targeted in this work, or if some approaches targeted only a subset of peptides. For example, did the parametric design approach fail to find any binders to SCT, Bid, or Bim, or where these peptides not targeted with this approach?

We targeted different sets of peptides using the different computational methods as they were developed at different times. In the case of the threading and RF Diffusion strategies we only included the highest affinity binders in the manuscript. The approaches used for each target are now listed in supplemental table 1, which is reproduced below.

Computational method	Targeted peptides
Parametric design	GCG, PTH and NPY
Hallucination	Bid
Inpainting	Parametrically designed binders for GCG, PTH and NPY
Threading design	PTH, NPY, GCG, SCT, GIP, GLP1, GLP2
Partial diffusion	Inpainted GCG and NPY binders
Unconditional diffusion	PTH and Bim
RFdiffusion_flex	PYY, NPY, GCG, Puma and PTH.

Fig 1b and c are vague and could be improved if the actual bundle topology that is being explored is depicted. The cartoon is too imprecise and potentially misleading. Please show the data for the “pilot experiments” mentioned in the supplement (Identification of weak binder hits from parametric designs in pilot experiment). I am not sure why these are called “pilot experiments” instead of “experiments”. What are the additional experiments that indicated > 100 nM binding affinities?

We have improved Fig 1b and c as suggested by the reviewer, and merged this with Figure 2 along with the data requested by the reviewer on the measured affinities (Supplementary Fig. S4).

Parametric design of groove-shaped scaffold library and use for binder design. “A supercoiling value was randomly selected from a biased distribution favoring more supercoiled scaffolds, given these scaffolds were more likely to fail in the subsequent looping step.” Please explain. This section would greatly benefit from a figure describing what the parameters and distributions are that are being sampled. The “adapted version of the miniprotein binder design pipeline” needs to be clearly described. The computational method as written in the current manuscript is not reproducible. How were the initial parametric binders designed? Was any backbone flexibility allowed during the design with peptide?

We have revised this section to explain the design steps in more detail, and provided figures to illustrate the parameters and distributions sampled (Supplementary Fig. S1 and Fig. S2). We also include links to the associated codebases in the Computational Methods section.

Design of BIM peptide binders. This section is inadequately described. “External potentials were used to promote interactions between the binder and target - specifically, the radius of gyration of the complex was minimized.” How? The authors need to significantly elaborate on the exposition of the methods.

Following reviewer suggestion, we have added a section describing the use of auxiliary potentials in the Methods section of the paper, and included references to the accompanying paper and code which describe their use in further detail.

It is now common practice to deposit plasmids to addgene for distribution and reproducibility. I do not think the Baker lab has deposited the petcon3 plasmid, which they commonly use for yeast display. I urge the authors to deposit the plasmid to addgene or publish the plasmid map.

We thank the reviewer for pointing this out. The vector used for yeast surface display experiments has been previously deposited in Addgene with the name pETCON-HtsptLB12v.3 (Plasmid #45121). Even though the deposited vector contains an insert, the plasmid backbone is identical to the pETcon3 vector we used in this manuscript. For inserting our design proteins we digested this plasmid with the NdeI and XhoI restriction enzymes. This is now clarified in the methods section.

“There is considerable interest in their sensitive and specific quantification, which currently relies on antibodies that require substantial resources to generate, can be difficult to produce with high affinity, and often have less-than-desirable stability and reproducibility⁵.” Could the authors cite some papers other than from the Baker lab to qualify the “considerable interest” mentioned here?

Following reviewer suggestion, we have now incorporated additional references for this in the main text. They are listed below:

Hocher, B. et al. Measuring parathyroid hormone (PTH) in patients with oxidative stress--do we need a fourth generation parathyroid hormone assay? PLoS One 7, e40242 (2012).

Shackman, J. G., Reid, K. R., Dugan, C. E. & Kennedy, R. T. Dynamic monitoring of glucagon secretion from living cells on a microfluidic chip. Anal. Bioanal. Chem. 402, 2797–2803 (2012).

Baker, M. Reproducibility crisis: Blame it on the antibodies. Nature 521, 274–276 (2015).

Bradbury, A. & Plückthun, A. Reproducibility: Standardize antibodies used in research. Nature 518, 27–29 (2015).

Bailly, M. et al. Predicting Antibody Developability Profiles Through Early Stage Discovery Screening. MAbs 12, 1743053 (2020).

Saper, C. B. A guide to the perplexed on the specificity of antibodies. J. Histochem. Cytochem. 57, 1–5 (2009).

Le Basle, Y., Chennell, P., Tokhadze, N., Astier, A. & Sautou, V. Physicochemical Stability of Monoclonal Antibodies: A Review. J. Pharm. Sci. 109, 169–190 (2020).

Lee, J. H., Yin, R., Ofek, G. & Pierce, B. G. Structural Features of Antibody-Peptide Recognition. *Front. Immunol.* 13, 910367 (2022).

Figure 2c. The hallucinated protein trajectory does not appear to be for the same protein shown to the left of the plot, which appears to be all helical. The trajectory has beta sheet. The binding isotherm only has one datapoint in the transition from fully unbound to fully bound. As such, the dissociation constant is sensitive to small errors in the single point that describes the transition. It would be better to measure more data points around this region in order to increase the confidence in the reported K_d value.

Unfortunately, for the purposes of memory consumption, we do not typically save the original trajectories, and do not have the precise trajectory that led to the binder shown in the figure. To prevent confusion, we have removed the trajectory from the main figure and instead included it as an example trajectory in Figure S9a. The example trajectory contains beta sheets as it is representative of many of the designs tested. If the reviewer particularly desires however, we would be happy to replace it with an all helical trajectory. It is correct that with only one point in the titration, we can conclude that the affinity is strong, but cannot extract an accurate dissociation constant. We have carried out additional fluorescence polarization titrations to accurately determine the affinity for the most promising Bid hallucination binder, which displays a stronger K_d than the ITC originally suggested. This data has been added to Fig 1e.

Please describe the yeast display protocol more thoroughly. Were the eblocks transformed into different cultures, were they pooled, etc? Were full eblocks ordered for each of the 192 designs?

eBlock fragments encoding our peptide binders were individually transformed into digested pETcon3 using the protocol outlined in the Methods section. For binding screening, we utilized the 96-well compatible autosampler in the Attune NxT Flow Cytometer (Thermo Fisher Scientific), which allowed for individual screening of yeast cells harboring each unique design. This is now clarified in the methods section.

The authors screened 12 designs for each target to evaluate the parametric design approach. They then screen 192 designs for a single target (PTH) using additional inpainting. They found a tighter binder. Does this necessarily mean that inpainting is able to produce better binders? I agree that additional buried surface area should lead to higher affinity, but there is likely a correlation between finding a tighter binder and testing more sequences per target. How much does inpainting matter here? Could the authors have just created longer parametrically generated bundles and/or simply screened more sequences experimentally?

We performed inpainting on a single parametrically designed binder for PTH, GCG, and NPY, which exhibited the highest binding signals in the NanoBiT assay. This resulted in the generation of 192 derivatives (in the case of the PTH binder) with improved lengths and shapes to better accommodate the peptide, allowing for affinity maturation without the need for a custom scaffold library specific to the target.

To investigate the contribution of ProteinMPNN to the increased binding affinity, we obtained 12 ProteinMPNN-redesigned sequences for the original PTH parametrically designed binder, without extending the binding interface or changing the backbone structure. Experimental testing revealed that ProteinMPNN redesign solely on the original binder did not lead to an increase in affinity, as shown in Supplementary Figure S5. We of course do not preclude the possibility that with more sampling, ProteinMPNN might be able to find a high affinity sequence, but empirically, we have found here that with the additional Inpainting step, we more readily find higher affinity binders, and rationalize this by the additional contact area made by these designs, as 44 out of the 192 designs tested showed binding against PTH in initial yeast display screening.

“We started from a library of scaffolds that contained single helices bound by pseudorepetitive helical scaffolds.” Where did this library come from? More detail is required, as this is otherwise not reproducible.

We would like to clarify that the library of scaffolds utilized in our study has been extensively described in a separate publication by our laboratory, as detailed in Praetorius et al. 2023. This previous work provides comprehensive information on the design, construction, and characterization of the scaffold library, including its composition and structural features. The library will be made available upon reasonable request.

“The binders were then redesigned in the presence of the threaded target sequence with ProteinMPNN and the complex was predicted with AF2 (with initial guess) and filtered on AF2 and Rosetta metrics.” Describe initial guess in the supplement, as well as each of the metrics used for filtering.

We now provide Jupyter notebooks for the entire threading and redesign pipeline. The AF2 initial guess approach is described exactly in Bennett et al. 2023 and in the associated github repository (https://github.com/nrbennet/dl_binder_design)

“Following size exclusion chromatography (SEC) purification of the monomer fraction...” Where is the SEC data? How much of these designs is not monomer? Does the collected monomer fraction re-equilibrate to higher order oligomerization state? This is important since this effects stoichiometry of binding and therefore could impact the reported binding affinity.

We appreciate the reviewer's comment and acknowledge that some of our designs displayed additional higher order peaks during size exclusion chromatography (SEC) purification. However, it is important to note that for conducting binding experiments, we specifically collected protein fractions corresponding to the highest abundance monodisperse peak to ensure the accuracy of our results. In response to the reviewer's suggestion, we now include SEC traces for either the initial purification or reinjections of SEC purified designs shown in the manuscript in figure S13.

Fig S4b, the chromatogram traces appear to be cut off prematurely. Please indicate which

elution time corresponds to the expected MW.

To clarify expected elution times we now include published elution times for proteins of known MW in Fig. S9b.

“The average contact molecular surface for the partially diffused GCG binders and NPY increased by 33% and 29% respectively compared to the starting models, and the Rosetta ddG improved by 29% and 21% (Fig. S5a, S5b).” What is this metric for the designs that experimentally bound their target? For the highest affinity binders?

The average contact molecular surface (CMS) and Rosetta ddG values were computed for the NPY binders using AF2 models and for the GCG binders using the recently solved crystal structures before and after partial diffusion. For the NPY inpainted binder, the CMS value was initially 506, which increased to 713.4 after the partial diffusion process. Similarly, the ddG value improved from -63.4 to -86.8. In the case of the GCG inpainted binder, the CMS increased from 421.46 to 521.75 after partial diffusion. The ddG value slightly improved from -26.017 to -27.79 in the partially diffused crystal structure.

The authors note that the “human”- vs “machine”- based solutions to this design problem were largely similar. Perhaps this similarity is not so surprising, given that there are many examples of helical bundle proteins in the PDB. Designers are quite familiar with helical bundles, as are the “machines.” RFdiffusion is highly biased to generate small helical proteins. The application mentioned here would be far more impressive to achieve on peptides with little secondary structure, but I imagine that RFdiffusion and other deep-learning methods will struggle with this task due to the much sparser training data. The authors should discuss these limitations or consider showing that the approach could be feasible for arbitrary peptides.

As noted above, beyond the scope of the paper our approach has been used to make binders for a number of amyloidogenic peptides and intrinsically disordered peptides. While these results go well beyond the context and scope of this work, we have included preliminary results in an accompanying slide deck and Appendix at the end of this document.

What do the authors mean by “wrong positioning of the antigen-binding site during sensor immobilization” and how does their approach address this?

We have revised to remove this sentence as we didn’t specifically try to address this problem during our design strategy.

There are no structures of apo or holo proteins. How do the authors know the designs are atomically accurate?

We successfully determined the crystal structures of the inpainted and partially diffused GCG binders, along with our picomolar Bim binder which are now included in Figures 2c and 3d.

Were any mutational studies performed to determine effects on binding affinity?

We did not perform any mutagenesis of the binders.

Fig 5a. This schema is confusing and mislabeled in the figure legend (There is no red color in the image.). Please update to clearly show a labeled construct. Where is the peptide, where is the designed binder?

We thank the reviewers for the feedback and have reworked this figure (now 4a) in order to increase clarity.

Fig 5b, shouldn't the emission saturate at some concentration? Is the affinity of the cage so high for the fused binder protein that the peptide cannot effectively outcompete it despite its pM affinity? Perhaps the authors could either run this experiment to saturation or provide an analysis for why the fluorescence does not saturate at 10 microM concentration of peptide.

We have run the proposed experiment using saturating concentrations of peptide, this data is now included in Figure 4b.

Fig S3, what is contact probability? Where are the scripts that ran these design trajectories?

We thank the reviewer for the comment, and have now included greater detail about the Hallucination approach, in the Computational methods section. We acknowledge that the hallucination code mentioned in the manuscript has indeed been deprecated and replaced with the more efficient RF Diffusion method, which provides a significant improvement in speed, being approximately 3-4 orders of magnitude faster. To ensure reproducibility and enable others to replicate our findings, we are able to make the legacy versions of the code available upon publication if the reviewers are concerned with reproducing the work.

"The plasma samples used were de-identified leftover clinical samples obtained from the clinical laboratories at the University of Washington Medical Center." What is a "leftover" clinical sample? Please indicate why PTH needed to be supplemented in the clinical plasma samples. Is it because the physiological levels (pg/ml) are too low for measurable enrichment by the peptide binder?

Leftover clinical samples are specimens from patients that are ready to be disposed of. The Research Testing Service is able to save these specimens before they are thrown in the trash, remove patient identifiers, and provide them to investigators. Specimens were supplemented with exogenous parathyroid hormone because the concentration of PTH was too low to be detectable by LC-MS/MS.

Please comment on if the majority of these designs needed to be co-expressed with the target peptide to prevent aggregation and if this presents a challenge to using these for clinical diagnostic purposes. Only the Rfdiffusion-derived proteins were commented to express

without need for co-expression of target peptide.

All designs in the paper solubly express and were purified without their target peptides.

Fig 4c. What is the stoichiometry of binding? The data looks like it might deviate from the expected 1:1 stoichiometry. This might affect the value of the dissociation constant.

We reran the binding titration of this design and incubated for longer to allow better equilibration of off rates. The new plot can be seen in Fig. 3b.

Where are the SEC chromatograms of the RFDiffusion-derived binders showing that they are monomeric?

In response to the reviewer's suggestion, we now include SEC traces for the RFDiffusion-derived binders. The RF Diffusion-derived binders are in Supplemental Fig. 13e-h.

Referee #3:

This is a companion paper to the RFDiffusion paper, as such my concerns about reproducibility of the model used in the aforementioned paper also apply to this work.

We have now provided detailed descriptions of the key computational methods in the revised manuscripts along with more detailed characterization of the designed proteins.

This work is concerned with designing binders to small helical peptides. As the authors point out, this is challenging for traditional methods as the binding interfaces for peptides are smaller than in design of binders for other targets, also peptides are often unstructured in isolation. This makes this problem well suited for machine learning methods, which have proven much more successful at predicting protein complex structures in cases where the proteins are only structured upon binding.

From my perspective this is a solid application work following what is a fairly well trodden path by now: take the latest structure generation model, redesign the sequence with ProteinMPNN and then filter out structures based on AlphaFold2 predictions. I am not surprised that this works well given the many similar successes of this approach in the past; the novelty here is using RFDiffusion as the generative model..

We thank the reviewer for sharing the encouraging perspective and would like to highlight the interesting diversity of methods that obtained high affinity binders in this work, the new structural and comparative analysis of determinants of high affinity, and the experiments showing their utility in diagnostic applications.

The experimental evaluation here looks very solid to me. Overall to me this looks like a good application paper; however I am not an expert on the biology part and as such can not really comment on it.

Alexander Pritzel

[REDACTED]

Reviewer Reports on the First Revision:

Referees' comments:

Referee #1:

The authors have satisfactorily answered all my concerns.

Referee #2:

The authors have made many welcome improvements to the manuscript. Re: my comment that the problem domain might be too easy, the authors point to the apparent lack of progress in the field since the late 90s, and the point is well taken.

I do think the authors still need to clarify a few points before publication (see comments below), especially regarding the use of FastRelax. The authors stress the point that subtle changes in the backbone were necessary for partial diffusion to find better interactions with the peptide. The authors used FastRelax, which also makes changes to the backbone, so it is hard for me to deconvolute the relative contributions of FastRelax vs partial diffusion as applied here. Even the diffusion-from-pure-noise protocol used FastRelax, which is perplexing to me. The explanation might be simple, but this point should be expressed more clearly in the paper.

It is nice to see new experiments that compared the performance of proteinMPNN on one of the original parametric designs.

I would like to see the hallucination protocol published since the authors report using it to generate binders, even if it is slower than RFdiffusion.

In the figures, coloring designs by AF2 pLDDT might be clearer, at least in the supplement, so that the reader can know what part of the peptide was confidently predicted in the designs. Only the half the GCG peptide was confidently predicted by AF2 in the best designed binders (Fig. 2c), but I only know this because I folded the sequences myself using AF2. I do not have access to the design models from RFdiffusion or the crystal structures. I would have liked to compare the structures and models. In particular the discussion around the Phe and Ile mutation seems very subtle indeed and the figure is not clear. If a Phe is poorly packed in the original design, what makes the Ile better packed in the partially diffused design? I hope the authors can clarify this in an updated figure. The predictions of the protein structures allowed me to partly assess the authors' claim "A 0.4 Å shift towards the target in the binder backbone enables an Ile to fit into a pocket previously occupied by a poorly packed Phe sidechain". It appears that an Ile could have substituted for the Phe without any backbone shifts, so the reason for the tighter binding is perhaps not so simple. Indeed there are many core mutations in this region (e.g., the Ser to Tyr mutation qualitatively seems more perturbative than Phe to Ile).

The authors also do not indicate if any design redundancy was considered in filtering sequences for

experimental testing. Could the authors please point out how diverse the designs were and how they dealt with redundancy in the designs (clustered by Tm score as well as their seq similarity)? I believe the sequences of the GCG inpainted vs partially diffused designs are about 34% similar, which is lower than I might have anticipated, suggesting redundancy in sequence is likely not an issue, but some clarification would be good. Discussion around the diversity of the sequences/structures would bolster the argument that the method is robust. (Otherwise it is trivial to say half of all the designs tested bound if they only differ by a single residue, for example. This is of course an extreme example just to make my point clear.)

In the rebuttal letter, the authors write “Specimens were supplemented with exogenous parathyroid hormone because the concentration of PTH was too low to be detectable by LC- MS/MS.” This should be more clearly spelled out in the paper, since in the paper the authors claim “Our MS based detection of peptides present at very low abundance in sera following enrichment using the designed binders could provide a general route forward for serological detection of a wide range of disease associated peptide biomarkers.” In the paper, the authors write that the plasma was spiked but do not describe why. Clarification here is important because the proposed method is for clinical detection, yet these are clinical samples and needed to be spiked, suggesting that either the binders were insufficient for the task or the clinical samples were not representative of normal samples that would have higher concentrations of PTH (although I believe the concentration of PTH in plasma is usually quite low).

It is very nice to see the new finetuned model of RFdiffusion_flex. In the rebuttal document and Suppl. Table 1, the authors show that RFdiffusion_flex was used to target other peptides in addition to PYY. Only PYY is mentioned in the paper. If the method failed to bind other peptide targets perhaps this should be mentioned at least in the supplement.

The time between the original SEC and reinjection SEC should be indicated in the legend of Suppl. Fig. 13. There does look to be some re-equilibration to an aggregated state for a few of the designs. The original SEC traces are not shown so it is difficult to assess if the peak corresponding to the higher order oligomer would continue to grow with time or if equilibration has been reached.

“The resulting library of backbones were sequence designed using ProteinMPNN and FastRelax, followed by AF2+initial guess9. The resulting libraries were filtered on AF2 pAE, pLDDT, RMSD to the design model, RMSD of the monomer to the binder model (without the peptide), and Rosetta ddg. The precise values used for filtering were chosen to reduce the set down to 96 designs for each target.”

The authors do not mention in the main text that they also use Rosetta FastRelax when designing the partially diffused binders. How much is FastRelax changing the backbone vs partial diffusion? (I assume FastRelax is altering the backbone coordinates.) Naively, I would have assumed that FastRelax would be unnecessary. Please indicate why it was used.

The equation in the supplement for fitting the dissociation constant appears to be incomplete. Perhaps it was a file rendering error upon submission. But the authors should make sure to update it with the correct formula. My file just shows $\text{Signal} = \text{baseline} + \text{amplitudeABconc}$.

The Kds in Fig. 2b are <0.5 nM and 1.3 nM; this would be a factor of 3 difference and not a factor of 10 for more specific binding, which the authors claim in the text.

“Initial sequence sampling: In line with Wicky et al., the initial binder sequence was sampled randomly, with amino acids probabilities corresponding to background amino acid frequencies in BLOSUM62 [cite].” The authors might want to fill in the citation.

Referee #3:

Overall I think the authors addressed all my comments and I think the work should be published.

Author Rebuttals to First Revision:

Referee #1:

The authors have satisfactorily answered all my concerns.

We thank the reviewer for recognizing our efforts to address the concerns raised, and we appreciate the time and attention you dedicated to our manuscript.

Referee #2:

The authors have made many welcome improvements to the manuscript. Re: my comment that the problem domain might be too easy, the authors point to the apparent lack of progress in the field since the late 90s, and the point is well taken.

We thank the reviewer for acknowledging the improvements we have made to our manuscript based on the thorough feedback.

I do think the authors still need to clarify a few points before publication (see comments below), especially regarding the use of FastRelax. The authors stress the point that subtle changes in the backbone were necessary for partial diffusion to find better interactions with the peptide. The authors used FastRelax, which also makes changes to the backbone, so it is hard for me to deconvolute the relative contributions of FastRelax vs partial diffusion as applied here. Even the diffusion-from-pure-noise protocol used FastRelax, which is perplexing to me. The explanation might be simple, but this point should be expressed more clearly in the paper.

We thank the reviewer for suggesting we clarify our use of FastRelax in sequence design. In the accompanying paper, (Watson et al 2023), we have shown that FastRelax is not systematically helpful in designing binders but rather, can occasionally rescue sequences coming directly from ProteinMPNN. Given the low computational cost of running FastRelax (a few CPU minutes per design), we deemed it helpful to do sequence design both with and without FastRelax. To better deconvolute the relative contributions of FastRelax vs partial diffusion, we have added a new Supplemental Figure and discussion (Fig. S13) showing that using FastRelax marginally improves *in silico* metrics of binding but partial diffusion enables significant improvements in binding even without it.

It is nice to see new experiments that compared the performance of proteinMPNN on one of the original parametric designs.

I would like to see the hallucination protocol published since the authors report using it to generate binders, even if it is slower than RFDiffusion.

Code for the hallucination will be published. We need a little bit of time to prepare this for a smooth public release, but will prepare this over the next couple of weeks and will append it to the paper during the proofs stage.

In the figures, coloring designs by AF2 pLDDT might be clearer, at least in the supplement, so that the reader can know what part of the peptide was confidently predicted in the designs. Only the half the GCG peptide was confidently predicted by AF2 in the best designed binders (Fig. 2c), but I only know this because I folded the sequences myself using AF2. I do not have access to the design models from RFDiffusion or the crystal structures. I would have liked to compare the structures and models.

We have included the design models and crystal structures as a supplementary zip for comparison purposes. We have also added a new supplementary figure (Fig. S14) which colors the AF2 models of the GCG inpainted and partially diffused binders by pLDDT.

In particular the discussion around the Phe and Ile mutation seems very subtle indeed and the figure is not clear. If a Phe is poorly packed in the original design, what makes the Ile better packed in the partially diffused design? I hope the authors can clarify this in an updated figure. The predictions of the protein structures allowed me to partly assess the authors' claim "A 0.4 Å shift towards the target in the binder backbone enables an Ile to fit into a pocket previously occupied by a poorly packed Phe sidechain". It appears that an Ile could have substituted for the Phe without any backbone shifts, so the reason for the tighter binding is perhaps not so simple. Indeed there are many core mutations in this region (e.g., the Ser to Tyr mutation qualitatively seems more perturbative than Phe to Ile).

We agree with the reviewer that this was not as clear as it could have been, and have tried to improve the presentation. To better illustrate the structural differences, we re-rendered the two binders in Fig. 2e aligned by the well structured and confidently predicted GCG C-terminal residues which were in a nearly identical conformation in both structures. Aligning in this manner emphasizes the differences in the backbone position, with a 2.7 Å shift at the Phe/Ile mutation site and a 3.6 Å shift at the Ser/Tyr site mentioned by the reviewer. We now highlight both of these positions as substitutions likely enabled by the change in backbone position, while taking care not to speculate extensively about any biophysical improvements resulting from the mutations.

The authors also do not indicate if any design redundancy was considered in filtering sequences for experimental testing. Could the authors please point out how diverse the designs were and how they dealt with redundancy in the designs (clustered by T_m score as well as their seq

similarity)? I believe the sequences of the GCG inprinted vs partially diffused designs are about 34% similar, which is lower than I might have anticipated, suggesting redundancy in sequence is likely not an issue, but some clarification would be good. Discussion around the diversity of the sequences/structures would bolster the argument that the method is robust. (Otherwise it is trivial to say half of all the designs tested bound if they only differ by a single residue, for example. This is of course an extreme example just to make my point clear.)

We thank the reviewer for suggesting we highlight the diversity of our sequences, especially for the partially diffused set. To show that our method is robust, and not generating minor variants of a single successful design, we have included a sequence logo and a sequence identity heatmap of the experimentally tested designs for both GCG and NPY, which is now included in the supplementary Figure S12.

In the rebuttal letter, the authors write “Specimens were supplemented with exogenous parathyroid hormone because the concentration of PTH was too low to be detectable by LC-MS/MS.” This should be more clearly spelled out in the paper, since in the paper the authors claim “Our MS based detection of peptides present at very low abundance in sera following enrichment using the designed binders could provide a general route forward for serological detection of a wide range of disease associated peptide biomarkers.” In the paper, the authors write that the plasma was spiked but do not describe why. Clarification here is important because the proposed method is for clinical detection, yet these are clinical samples and needed to be spiked, suggesting that either the binders were insufficient for the task or the clinical samples were not representative of normal samples that would have higher concentrations of PTH (although I believe the concentration of PTH in plasma is usually quite low).

We appreciate the reviewer's suggestion regarding the need for clarification in the paper. We have incorporated this revision into the main text of the manuscript.

It is very nice to see the new finetuned model of RFdiffusion_flex. In the rebuttal document and Suppl. Table 1, the authors show that RFdiffusion_flex was used to target other peptides in addition to PYY. Only PYY is mentioned in the paper. If the method failed to bind other peptide targets perhaps this should be mentioned at least in the supplement.

We appreciate the reviewer's inquiry regarding our selection of binders for detailed characterization. Initially, our focus was on the exhaustive characterization of PYY binders, given that we had successfully obtained high-affinity binders for the other targets through alternative computational methods. In response to the reviewer's feedback, we have included representative binding data for GCG and PTH protein binders designed using RFdiffusion_flex in Supplementary Figure S16.

The time between the original SEC and reinjection SEC should be indicated in the legend of Suppl. Fig. 13. There does look to be some re-equilibration to an aggregated state for a few of the designs. The original SEC traces are not shown so it is difficult to assess if the peak

corresponding to the higher order oligomer would continue to grow with time or if equilibration has been reached.

We have incorporated the time between the original SEC and reinjection in the legend of what is now Supplemental Figure S18.

“The resulting library of backbones were sequence designed using ProteinMPNN and FastRelax, followed by AF2+initial guess9. The resulting libraries were filtered on AF2 pAE, pLDDT, RMSD to the design model, RMSD of the monomer to the binder model (without the peptide), and Rosetta ddg. The precise values used for filtering were chosen to reduce the set down to 96 designs for each target.”

The authors do not mention in the main text that they also use Rosetta FastRelax when designing the partially diffused binders. How much is FastRelax changing the backbone vs partial diffusion? (I assume FastRelax is altering the backbone coordinates.) Naively, I would have assumed that FastRelax would be unnecessary. Please indicate why it was used.

We now show in Supplemental Figure S13 that partial diffusion is necessary for significant improvement in binding metrics (interaction pAE). FastRelax + MPNN alone only results in marginal improvements.

The equation in the supplement for fitting the dissociation constant appears to be incomplete. Perhaps it was a file rendering error upon submission. But the authors should make sure to update it with the correct formula. My file just shows $\text{Signal} = \text{baseline} + \text{amplitude} \times \text{ABconc}$.

We thank the reviewers for catching this issue in the manuscript and have fixed this in the updated version. The corrected full equation now reads

$$\text{Signal}_i = \text{Baseline} + \text{Amplitude} \frac{AB_{conc}([A_{tot}], [B_{tot}], K_d)}{[B_{tot}]}$$

The Kds in Fig. 2b are <0.5 nM and 1.3 nM; this would be a factor of 3 difference and not a factor of 10 for more specific binding, which the authors claim in the text.

We thank the reviewers for identifying this issue in the manuscript. We have addressed and resolved it in the updated version of the paper.

“Initial sequence sampling: In line with Wicky et al., the initial binder sequence was sampled randomly, with amino acids probabilities corresponding to background amino acid frequencies in BLOSUM62 [cite].” The authors might want to fill in the citation.

We have updated the citation to reference the intended original BLOSUM62 paper (Henikoff and Henikoff, *PNAS* 1992).

Referee #3:

Overall I think the authors addressed all my comments and I think the work should be published.

We thank the reviewer for the positive evaluation and for recognizing our efforts in addressing the comments and appreciate the careful attention and time you committed to our manuscript.